# A Roadmap for NF-ISAC in 6G: A Comprehensive Overview and Tutorial

**DOI:** 10.3390/e26090773

**Published:** 2024-09-10

**Authors:** Azar Hakimi, Diluka Galappaththige, Chintha Tellambura

**Affiliations:** Department of Electrical and Computer Engineering, University of Alberta, Edmonton, AB T6G 1H9, Canada; hakimina@ualberta.ca (A.H.); diluka.lg@ualberta.ca (D.G.)

**Keywords:** near-field communications, integrated sensing and communication, channel modeling, optimization problem

## Abstract

Near-field (NF) integrated sensing and communication (ISAC) has the potential to revolutionize future wireless networks. It enables simultaneous communication and sensing operations on the same radio frequency (RF) resources using a shared hardware platform, maximizing resource utilization. NF-ISAC systems can improve communication and sensing performance compared to traditional far-field (FF) ISAC systems by exploiting the unique propagation characteristics of NF spherical waves with an additional distance dimension. Despite its potential, NF-ISAC research is still in its early stages, and a comprehensive survey of the technology is lacking. This paper systematically explores NF-ISAC technology, providing an in-depth analysis of both NF and FF systems, their applicability in various scenarios, and different channel models. It highlights the advantages and philosophies of ISAC, examining both narrow-band and wide-band NF-ISAC systems. Case studies and simulations offer deeper insights into NF-ISAC design philosophies. Additionally, the paper reviews the existing NF-ISAC literature, methodologies, potentials, and conclusions, and discusses future research areas, challenges, and applications.

## 1. Introduction

The sixth generation (6G) of cellular networks is anticipated to be available in early 2030, and the research journey has already commenced [1]. The advent of 6G is poised to contribute to the establishment of a human-friendly intelligent society. Consequently, the targets set for 6G excel those of previous generations. Therefore, it is set to enable new applications, such as extended reality (XR), holographic communication, pervasive intelligence, Internet of Things (IoT), autonomous driving in vehicle-to-everything (V2X), smart traffic control, smart homes, unmanned aerial vehicles (UAVs), and factory automation [2]. Meanwhile, the International Telecommunications Union (ITU) has announced its aspirations for 6G, including a timeline, future technology trends, recommended frameworks, and so on [3]. This marks the official start of the journey towards 6G standardization. They defined six major 6G use cases, including three extended fifth generation (5G) application scenarios, namely, immersive enhanced mobile broadband (eMBB+), massive machine type communication (mMTC+), hyper ultra-reliable and low-latency communication (URLLC+), and three new items that will thrive in the new era of 6G, namely, integrated sensing and communication (ISAC), integrated artificial intelligence (AI) and communication, and ubiquitous connectivity.

To facilitate these application requirements, the data rate and connectivity of 6G must surpass that of the 5G by a factor of 100 and 10 times, respectively. This results in a minimum peak rate of 1 Tbps (terabit per second) and connectivity of 10^7^ devices per square kilometer [4]. Nevertheless, applications requiring such high data rates and connectivity cannot be adequately supported within a saturated sub-6 GHz bandwidth. On the other hand, these applications also necessitate ultra-reliable, low-latency, and high-precision sensing in addition to communication [5,6,7]. In this regard, numerous prospective 6G candidate technologies have emerged, including extremely large-scale multiple-input multiple-output (XL-MIMO) or extremely large-scale antenna arrays (ELAAs), ISAC, and millimeter wave (mmWave) and terahertz (THz) communication [8,9,10,11].

Integrating emerging ISAC systems with ELAAs operating in higher frequency bands leads to a wireless paradigm known as near-field (NF)-ISAC [12,13,14]. In particular, large antenna arrays typically provide a high spatial resolution, needed for angle-of-arrival (AoA) estimation and angular beamforming [12,13,14]. In addition, when the aperture of antenna arrays is extremely large and is combined with extremely high operating frequencies, the inevitable NF effect occurs. The NF and far-field (FF) are two intrinsic electromagnetic (EM) zones associated with antenna arrays (Section 2). Current wireless networks primarily operate in the FF region due to small antenna array apertures and lower operating frequencies. However, with the emergence of ELAAs and high-frequency bands, the NF zone extends to tens or even hundreds of meters, gaining prominence in sensing and communication applications [2]. While the NF effect introduces complexity to communication and sensing channels, it also presents new opportunities for NF-ISAC through the use of ELAAs [2,12,13,14]. Despite this potential, NF-ISAC remains unexplored, which has motivated a thorough investigation of its capabilities.

### Motivation and Contribution

Although several surveys and tutorials have addressed various aspects of ISAC and NF communication separately [2,6,9,10,15,16], there are only a few works specifically focused on NF- ISAC [13,14,17].

In particular, ref. [13] discusses the opportunities and challenges of NF communication and sensing compared to FF counterparts. It highlights the advantages of NF, NF-ISAC designs, associated challenges, and existing solutions while suggesting future directions. Reference [14] compares NF and FF characteristics, emphasizing NF’s advantages in communication and sensing. The included simulations demonstrate the potential for power savings in NF-ISAC systems. In [17], the fundamentals, challenges, and opportunities of NF wide-band ISAC systems are discussed. The study emphasizes the implications of significant angular-delay correlation in NF wide-band systems on communication and sensing functions, such as signal multiplexing and range sensing.

These studies have inspired this work, recognizing the need for more comprehensive research. The contributions of this paper are summarized as follows:In contrast to previous survey papers [13,14,17], this study offers a comprehensive examination covering all facets of NF-ISAC systems. Specifically, it thoroughly analyzes NF and FF systems, exploring their respective applications in communication and sensing scenarios. Various channel model scenarios for NF and FF are presented. Additionally, the merits and philosophies of ISAC are explored, paving the way to investigate both narrow-band and wide-band systems within NF-ISAC.To gain deeper insights into this domain, an NF-ISAC integrated with a non-orthogonal multiple access (NOMA) communication system and a target sensing system is proposed. Three case studies are defined. Case 1 prioritizes the sensing system, aiming to solve a sensing signal-to-noise ratio (SNR) maximization problem while ensuring quality of service (QoS) for the communication system. Conversely, Case 2 focuses on optimizing the communication system by maximizing its sum rate while maintaining QoS for the sensing system. Case 3 introduces the sum-weighted rate of users’ communication and sensing rates.An extensive simulation is conducted to evaluate the performance of the proposed case studies. The proposed design is compared with three other benchmarks for deeper insights: far-field (FF), communication-only, and sensing-only schemes. Numerical results indicate that the proposed NF design outperforms the FF counterpart in communication and sensing, demonstrating superior efficiency and effectiveness.A thorough literature review of the existing works is conducted, exploring various studies, methodologies, and findings in NF-ISAC. Insightful conclusions are drawn from the review, focusing on potential future research directions and the challenges. By examining current trends and knowledge gaps, valuable insights are provided to guide future research endeavors.

This survey study distinguishes itself from earlier ones by focusing on all aspects of NF-ISAC systems. It covers the most up-to-date research progress not reported in previous surveys. By exploring recent breakthroughs in NF-ISAC research, the study aims to expand the knowledge base and encourage further investigation in this rapidly developing field. Ultimately, this survey is a valuable resource for scholars, practitioners, and enthusiasts of NF-ISAC wireless research.

The structure of this paper is outlined in Figure 1.

*Notation*: Vectors (lower-case) and matrices (upper-case) are bold. $A^{T}$, $A^{H}$, Tr(A), rank(A), and diag(A) denote transpose, Hermitian transpose, trace, rank, and diagonal operator. A⪰0 denotes a positive semidefinite matrix. $I_{M}$ is the M×M identity matrix. The Euclidean norm, expectation, and absolute value operators are denoted by ∥·∥, E[·], and |·|. CN(μ,R) is a complex Gaussian vector with mean μ and co-variance R. The operators ${\left(\;\cdot \;\right)}^{-1}$ and ⊗ denote the inversion of a matrix and the Kronecker product. $C^{M\times N}$ represents M×N complex matrices. Finally, M≜{1,…,M}, K≜{1,…,K}, L≜{1,…,L}, $K_{k}≜K\setminus \left\{k\right\}$, and $L_{l}≜L\setminus \left\{l\right\}$.

## 2. Near-Field for Communication and Sensing

The EM propagation environment of a radio frequency (RF) source, also known as a radiation source, can be separated into three zones (Figure 2): reactive NF, radiating NF, and radiating FF [2]. It is important to note that the relative distance between the transceivers distinguishes these regions rather than the precise location of the transmitter.

*Reactive NF:* The boundary separating the reactive near field (NF) from the radiating NF is called the Fresnel distance, calculated as $0.5\sqrt{D^{3}/\lambda }$, where *D* is the antenna aperture size and λ is the electromagnetic wavelength. Thus, the region within this distance (i.e., $r<0.5\sqrt{D^{3}/\lambda }$) from the transmitter is the reactive NF. In this region, the electric and magnetic field components are out of phase, causing the EM field’s energy to oscillate rather than being fully radiated. Evanescent waves, or non-propagating fields, dominate, rapidly diminishing with distance. Significant amplitude and phase nonlinearities occur across the transmitter antenna array [2].*Radiating NF:* The boundary between the NF and FF is calculated as $2D^{2}/\lambda$ and is referred to as the Rayleigh distance. Indeed, the radiating NF is situated in the region between the Fresnel and Rayleigh distances (e.g., $0.5\sqrt{D^{3}/\lambda }\leq r\leq 2D^{2}/\lambda$). The electric and magnetic fields are perpendicular and in phase, generating propagating waves. However, the fields have not yet developed into ordinary planar waves, and the angular field distribution is influenced by the distance between receivers, resulting in spherical wavefronts. Spherical waves, which have nonlinear phase changes across the antenna aperture and varying amplitude depending on transceiver distance, thus dominate radio propagation [2].*Radiating FF:* This region is beyond the Rayleigh distance, which surrounds the radiating NF region. In this region, the signal paths between each point on the transmitter and the receiver can be treated as parallel, i.e., the angular field distribution is almost independent of the distance between the receiver and transmitter. This generates planar wavefronts with linear phase fluctuations and radio propagation, with the planar waves having the slowest decay rate of all [2].

As the reactive NF region or Fresnel region is often small, i.e., a few wavelengths and evanescent waves decay exponentially with distance, this work focuses only on the radiating NF region. For the sake of brevity, it is called the NF region throughout the rest of the paper.

While NF propagation has always existed, previous generations of wireless technologies have primarily relied on FF propagation due to the small antenna array sizes and sub-6 GHz operating frequencies. However, advancements in ELAAs with hundreds or even thousands of antennas (typically more than 100) and the use of higher frequency bands, such as mmWave (10 GHz to 100 GHz) and THz (100 GHz to 10 THz), have fundamentally shifted the EM propagation environment from FF to NF. For example, an ELAA with a length of 7.4 m operating at 2.6 GHz has a Rayleigh distance of 950 m.

There is no strict distance threshold or boundary between the NF and FF zones; instead, the transition occurs gradually [2,12,14]. Thus, the boundary between NF and FF depends on the unique application scenarios. As a result, various metrics have been developed to describe the field boundary. The two most typically utilized criteria are the phase error and channel gain error perspectives [2].

*Phase error perspective:* If the phase difference is less than 22.5°, the wavefronts have minimal curvature and can be approximated as plane waves; otherwise, they retain the spherical wave [14,18]. Some relevant metrics are the Rayleigh distance, Fraunhofer condition, and extended Rayleigh distance for MIMO transceivers and reconfigurable intelligent surfaces (RIS). Among these, the Rayleigh distance, the most widely used metric, is proportional to the product of the carrier frequency (1/λ) and the square of the array aperture size ($D^{2}$), i.e., $2D^{2}/\lambda$. The Fraunhofer condition, which satisfies the Fraunhofer diffraction equation, simulates wave diffraction. For antennas bigger than a half-wavelength, the NF and FF are defined in terms of the Fraunhofer distance, i.e., $d_{F}=2D^{2}/\lambda$. The extended Rayleigh distance based on the phase difference defines the NF and FF between the transmitter and receiver with large antenna arrays. The NF region is defined as $r<4D_{t}D_{r}/\lambda$, where *r* is the distance of the first antenna at the receiver from the first antenna at the transmitter, and  $D_{t}$ and $D_{r}$ are the array aperture of the transmitter and the receiver, respectively. These distances primarily pertain to the field boundary near the main axis of the antenna aperture.*Channel gain error perspective:* This provides a more precise definition of the field boundary for off-axis locations. The Friis formula, which states that channel gain diminishes with the inverse of distance squared [19], applies in the FF region but not the NF region. Therefore, the FF region is defined as where the Friis formula can approximate the actual channel gain with tolerable error; otherwise, it is the NF region. In this case, the field boundary depends not only on aperture size and wavelength, but also on the angle of departure, angle of arrival, and the shape of the transmit antenna aperture.

Unlike FF planar wavefronts, which are predominantly influenced by the propagation angle, NF spherical waves introduce an additional distance dimension, which substantially impacts the shape and behavior of the wavefront. In particular, the FF steering vector points to a specific direction, whereas the NF focusing vector focuses on a specific location [14]. This emphasizes a critical distinction between FF beam steering and NF beam-focusing, resulting in substantial theoretical and technological shifts in NF communication and sensing compared to conventional FF counterparts.

Figure 2 illustrates waveform propagation in the NF and FF, showing the spherical wave in the NF and the planar wave in the FF, distinguishable by the Rayleigh distance.

### 2.1. NF Communication

The properties of NF spherical waves can increase channel degrees of freedom (DoFs) and boost system capacity [2,14]. In FF communication, the phase difference between signals received or transmitted by different antennas is linear, which is mainly governed by the wavefront’s angle of arrival or departure, as well as the antenna spacing [2,14]. Since the FF wavefront is planar, the phase difference between antennas varies linearly with their spatial position along the propagation direction, i.e., the angles between each transmitter antenna element and the receiver are almost identical in the FF region due to the large distance between them. This leads to a uniform phase shift across all antennas, and each link’s propagation distance grows linearly with the antenna index. This results in a single DoF in the product of the line of sight (LoS) and steering vector [2,14]. In contrast, the phase difference between the NF transceiver antennas is nonlinear and dependent on the distance between each antenna [2]. In NF spherical wavefronts, the variation in distances from the transmitter to each receiver antenna leads to varying phase shifts. Thus, each antenna has its propagation link distance information, resulting in enhanced DoF [2]. Moreover, traditional FF communication cannot distinguish users with the same angle, resulting in significant inter-user interference. However, NF communication can distinguish users with the same angle based on different distances and effectively mitigate inter-user interference owing to the nonlinear phases that relate to the distance of each antenna [2].

The additional distance dimension of spherical waves can mitigate multi-user interference. In particular, it helps to de-correlate the multi-user channel, pushing it closer to the optimal propagation condition. In contrast to FF communication beamforming, i.e.,  guiding beam energy in a specific direction, NF communication beamforming based on spherical wavefronts achieves a new function of beam-focusing, which focuses beam energy in a particular location/region [2,12,14]. This increases the received signal power at the intended user and removes interference from undesired users, allowing for massive connectivity. As the rank of the NF LoS MIMO channel exceeds one, the spatial multiplexing gain can potentially increase.

### 2.2. NF Sensing

Sensing requires both signal transmission and reception. In signal transmission, the focused and tightly confined beams in NF beam-focusing allow the signal energy to be focused on the target point with less signal spreading, resulting in increased signal power at the target. On the other hand, as NF spherical waves convey both distance (range) and angle information, NF reception provides selective signal capturing, allowing for targeted reception from specific transmitters while minimizing interference. It also reduces the need for distributed arrays and their synchronization [12,14]. NF beam-focusing improves the sensing SNR of echo signals, allowing for more precise estimation. Combining NF reception with advanced estimate techniques and spectrum search algorithms yields higher position estimation accuracy than the FF counterpart. For example, multiple antenna arrays with known spatial configurations allow for the indirect inference of distance information based on signal intensity, time of arrival, or phase discrepancies between antennas [14]. The additional feature of spherical wavefronts in the range provides improved recognition accuracy, facilitating applications such as human-activity recognition.

However, the distinct advantages of both NF communication and NF sensing are highly dependent on proper transmission architecture [14]. Conventional FF transmission designs, in particular, cannot be easily transferred to NF due to model mismatches, such as the channel model, resulting in significant performance loss. For instance, at close ranges, FF beams become divergent and broader, increasing user interference and angle estimation errors [14]. Consequently, existing FF beamforming strategies drastically reduce NF communication rates and sensing accuracy. Therefore, to fully realize the benefits of NF, specialized transmission designs and improved beamforming techniques that account for the features of NF channels are necessary [14].

## 3. Near-Field Channel Models

This section covers the foundations of the NF channel model, as illustrated in Figure 3. Conventional FF systems, with signal propagation distances greater than the Rayleigh distance, primarily use planner wave-based channel models. This leads to a uniform planar wave (UPW) with a linear phase and uniform power fluctuation of signals. However, NF systems must utilize spherical wave-based channel models to accurately capture NF characteristics, such as nonlinear phase variations [2]. In addition, the NF zone can be separated into uniform and non-uniform amplitude regions using the uniform power distance to account for signal amplitude variations. NF channel models may be generally classified into two categories based on Rayleigh and uniform power distances: (i) uniform spherical wave (USW) models, and (ii) non-uniform spherical wave (NUSW) models. These NF channel models vary in accuracy and attributes according to the involved assumptions.

NF channel models are further sub-classified as deterministic or statistical models that take scattering and small-scale fading into account [2]. In particular, deterministic models employ ray tracing, geometric optics, or EM wave propagation theories to precisely determine channel gain, mainly for LoS or channels with a finite number of paths. In contrast, statistical models incorporate the channel’s average behavior, fading effects, and time-varying properties, making them appropriate for defining rich-scattering environments [2]. On the other hand, transceiver antenna types such as spatially discrete (SPD) and continuous-aperture (CAP) antennas significantly influence NF channel models [2]. Several widely used NF channel models and their characteristics are reviewed next.

### 3.1. USW Model for SPD Antennas

Without loss of any generality, assume $M=2\hat{M}+1$ antenna transmitter and single-antenna receiver, where the transmitter antenna index is given by $m\in \{-\hat{M},\ldots ,\hat{M}\}$. The coordinates of the *m*-th element of the transmitter antenna array and the receiver are denoted as $s_{m}={\left[s_{m,x},s_{m,y},s_{m,z}\right]}^{T}$ for $m\in \{-\hat{M},\ldots ,\hat{M}\}$ and $r={\left[r_{x},r_{y},r_{z}\right]}^{T}$, respectively. It is further assumed that the central element of an antenna array is located in the origin of the coordinate system, i.e., $s_{0}={\left[0,0,0\right]}^{T}$. In addition, θ and ϕ are the azimuth and elevation angles of the receiver, respectively, relative to the x−z plane. The propagation distance between the *m*-th antenna element and the receiver is $r_{m}=∥r-s_{m}∥$ [2]. Therefore, the channel coefficient between the *m*-th transmit antenna element and the receiver is given by [2,9]:(1)$$\begin{array}{c} h_{m,\mathrm{NF}}=\beta_{m}e^{-j\frac{2\pi }{\lambda }r_{m}}, \end{array}$$
where $\beta_{m}$ represents the channel gain (amplitude) for the m-th link. Let $r=∥r-s_{0}∥$ be the propagation distance between the receiver and the central element of the transmit antenna array. Assuming *r* is greater than the uniform power distance, resulting in $\beta_{-\hat{M}}\approx \beta_{-\hat{M}+1}\approx \cdots \approx \beta_{\hat{M}}=\beta$, the NF LoS channel between the transmitter and receiver can be represented as:(2)$$\begin{array}{c} h_{\mathrm{NF}}^{\mathrm{LoS}}=\beta e^{-j\frac{2\pi }{\lambda }r}a\left(r\right), \end{array}$$
where a(r) is the NF array response vector, given as:(3)$$\begin{array}{c} a\left(r\right)={\left[e^{-j\frac{2\pi }{\lambda }\left(∥r-s_{-\hat{M}}∥-r\right)},\ldots ,e^{-j\frac{2\pi }{\lambda }\left(∥r-s_{\hat{M}}∥-r\right)}\right]}^{T}. \end{array}$$
As a result, the phase of the *m*-th entry of (Equation 3) is not a linear function of $s_{m}$.

In rich-scattering environments, the receiver receives signals that scatterers reflect through non-LoS (NLoS) channels. The randomness of these multi-path NLoS components also causes channels to be random. Thus, a statistical channel model is necessary. Let *L* be the total number of scatterers, $r_{l}$ the coordinate of the *l*-th scatterer, and ${\tilde{\beta }}_{l}$ the channel gain, including the random reflection coefficient of the *l*-th scatterer. Then, the NF multi-path channel can be modeled as:(4)$$\begin{array}{c} h_{\mathrm{NF}}=h_{\mathrm{NF}}^{\mathrm{LoS}}+\sum_{l\in L}\underset{\mathrm{NLoS}}{\underset{︸}{{\tilde{\beta }}_{l}a\left(r_{l}\right)}}, \end{array}$$
where the random phase of ${\tilde{\beta }}_{l}$ is assumed to be independent and identically distributed (i.i.d.) and uniformly distributed in (−π,π]. The NF array response vectors of two different antenna array geometries, uniform linear array (ULA) and uniform planar array (UPA), are discussed next to gain further insight.

*Uniform linear array:* A ULA is a one-dimensional linear antenna array with equal antenna spacing of *d*. The antenna array is placed in the x−y plane, with the origin of the coordinate system at the center of the ULA, resulting in $\phi =90^{\circ }$, i.e.,  the *z*-axis can be disregarded. The coordinates of the receiver and *m*-th ULA element are $r={\left[r\cos\left(\theta \right),r\sin\left(\theta \right)\right]}^{T}$ and $s_{m}={\left[md,0\right]}^{T}$. The propagation distance can be approximated as:
(5)$$\begin{array}{ccc} ∥r-s_{m}∥ & = & \sqrt{r^{2}+m^{2}d^{2}-2rmd\cos\left(\theta \right)}\\ & \overset{\left(a\right)}{\approx } & r-md\cos\left(\theta \right)+\frac{m^{2}d^{2}\sin^{2}\left(\theta \right)}{2r}, \end{array}$$
where the step (a) is obtained by using Fresnel approximation [18]. The *m*-th entry of the antenna array response for a ULA is thus given as:
(6)$$\begin{array}{c} {\left[a_{\mathrm{ULA}}\left(r,\theta \right)\right]}_{m}=e^{-j\frac{2\pi }{\lambda }\left(-md\cos\left(\theta \right)+\frac{m^{2}d^{2}\sin^{2}\left(\theta \right)}{2r}\right)}. \end{array}$$*Uniform planar array:* A UPA is a two-dimensional array of antennas uniformly arranged in a rectangular grid. The UPA is placed in the x−z plane, with the origin of the coordinate system at the center of the UPA. Assume $M=M_{x}\times M_{z}$ antenna elements, with $M_{x}=2{\hat{M}}_{x}+1$ and $M_{z}=2{\hat{M}}_{z}+1$, and $d_{x}$ and $d_{z}$ antenna spacings in the *x* and *z* directions, respectively. The coordinates of the receiver and (m,n)-th ULA element are $r={\left[r\cos\left(\theta \right)\sin\left(\phi \right),r\sin\left(\theta \right)\sin\left(\phi \right),r\cos\left(\phi \right)\right]}^{T}$ and $s_{m,n}={\left[md_{x},0,nd_{z}\right]}^{T}$, respectively, where $m\in \{-{\hat{M}}_{x},\ldots ,{\hat{M}}_{x}\}$ and $n\in \{-{\hat{M}}_{z},\ldots ,{\hat{M}}_{z}\}$. The propagation distance is then approximated as:
(7)$$\begin{array}{ccc} ∥r-s_{m}∥ & = & \sqrt{r^{2}+m^{2}d_{x}^{2}+n^{2}d_{z}^{2}-2rmd_{x}\cos\theta \sin\phi -2rnd_{z}\cos\phi }\\ & \overset{\left(b\right)}{\approx } & r-md_{x}\cos\theta \sin\phi +\frac{m^{2}d_{x}^{2}\left(1-\cos^{2}\theta \sin^{2}\phi \right)}{2r}\\ & & -nd_{z}\cos\phi +\frac{n^{2}d_{z}^{2}\sin\phi }{2r}, \end{array}$$
where the step (b) is computed via Fresnel approximation, assuming $d_{x}/r\ll 1$ and $d_{z}/r\ll 1$, and omitting the bi-linear term [18]. This approximation is adequate for the USW model [2]. By eliminating the constant phase, i.e., $e^{-j\frac{2\pi }{\lambda }r}$, the array response vector’s phase can be separated into two components: (i) $-md_{x}\cos\theta \sin\phi +\frac{m^{2}d_{x}^{2}\left(1-\cos^{2}\theta \sin^{2}\phi \right)}{2r}$ and (ii) $-nd_{z}\cos\phi +\frac{n^{2}d_{z}^{2}\sin\phi }{2r}$, which only depend on *m* and *n*. Thereby, the NF array response vector for a UPA can be given as:
(8)$$\begin{array}{c} a_{\mathrm{UPA}}\left(r,\theta ,\phi \right)=a_{x}\left(r,\theta ,\phi \right)⊗a_{z}\left(r,\theta ,\phi \right), \end{array}$$
where the *m*-th entry of $a_{x}\left(r,\theta ,\phi \right)$ and the *n*-th entry of $a_{z}\left(r,\theta ,\phi \right)$ are given as:
(9a)$$\begin{array}{cc} {\left[a_{x}\left(r,\theta ,\phi \right)\right]}_{m} & =e^{-j\frac{2\pi }{\lambda }\left(-md_{x}\cos\theta \sin\phi +\frac{m^{2}d_{x}^{2}(1-\cos^{2}\theta \sin^{2}\phi }{2r}\right)}, \end{array}$$
(9b)$$\begin{array}{cc} {\left[a_{z}\left(r,\theta ,\phi \right)\right]}_{n} & =e^{-j\frac{2\pi }{\lambda }\left(-nd_{z}\cos\phi +\frac{n^{2}d_{z}^{2}\sin\phi }{2r}\right)}. \end{array}$$

### 3.2. NUSW Model for SPD Antennas

When the propagation distance *r* is less than the uniform power distance, the channel gain changes are not negligible. The channel gains of various links are thus not uniform and must be estimated independently. In particular, the channel gain between the *m*-th transmit and the *n*-th receiver antenna elements can be calculated using the free-space path loss as $\beta_{mn}=1/\sqrt{4\pi ∥r_{n}-s_{m}{∥}^{2}}$. The respective channel coefficient of the NUSW model is thus given as [2]:(10)$$\begin{array}{c} h_{\mathrm{NF}}\left(s_{m},r_{n}\right)=\frac{1}{\sqrt{4\pi ∥r_{n}-s_{m}{∥}^{2}}}e^{-j\frac{2\pi }{\lambda }{∥r_{n}-s_{m}∥}^{2}}. \end{array}$$

### 3.3. Channel Characteristics

Modeling the NF propagation channel presents several unique challenges. For instance, NF systems operating in high-frequency bands encounter electromagnetic wavefronts with pronounced curvature and complex interactions with objects, leading to scattering, reflection, and diffraction behaviors that differ significantly from those in lower-frequency bands [20]. As a result, traditional models like Friis’ free space path loss, which are effective for conventional channel (e.g., FF) conditions, may not accurately capture the intricacies of NF propagation.

Theoretical modeling of the NF channel requires accounting for wavefront curvature, elevated path loss exponents, and the impact of small-scale phenomena, such as scattering from particles or surface roughness. Additionally, these models must consider heightened sensitivity to environmental factors, such as humidity and material properties, which become more significant at higher frequencies like mmWave and THz bands [20].

While advancements have been made in understanding and modeling NF propagation, existing models—particularly those developed for lower frequency bands, like the 3GPP TR36.873 for below 6 GHz—may not fully capture the NF behavior at higher frequencies. For example, in LoS conditions observed in urban micro (UMi) and urban macro (UMa) environments, the path loss at higher frequencies follows Friis’ free space path loss model. However, in NLoS conditions, a higher path loss slope (or path loss exponent) is observed, indicating that shadow fading, path loss, and delay spreads in NF high-frequency scenarios can differ significantly from conventional channel characteristics [21].

## 4. Integrated Sensing and Communications

This section briefly reviews ISAC fundamentals and recent developments before proceeding to the NF-ISAC networks.

The fundamental premise of ISAC revolves around the fusion of sensing and communication operations, leveraging the same hardware platform, shared spectrum, collaborative signal processing techniques, and even a unified control framework [15]. When conventional communication systems strive to achieve channel capacity, the goal is to introduce as much randomness into the signaling as possible [22]. Conversely, a certain degree of determinism in the wireless signal becomes imperative to enhance sensing performance. Consequently, the endeavor to concurrently implement both sensing and communication functions gives rise to a pivotal trade-off, succinctly encapsulating the essence of the random-deterministic dilemma, which is but one facet of the multifaceted conflicts of interest arising in the amalgamation of sensing and communication within a single system. It is precisely these intricate challenges and trade-offs that render ISAC research not only compelling but also imperative in advancing our understanding of integrated wireless systems.

Sensing plays a crucial role in 6G, particularly in applications such as localization [16], smart transportation [23], smart cities [24], and other location-based services. Currently, communication and sensing systems utilize separate spectrum resources. However, given the rapid growth of wireless services and mobile devices, using separate frequency bands may not be spectrum-friendly [25]. Nevertheless, the transition to higher frequency bands like mmWave and THz and super-massive MIMO technology results in sensing and communication sharing similar system architectures, channel characteristics, and signal processing methods [25]. Consequently, both sensing and communication can leverage the same equipment and waveform simultaneously, marking a new era for dual-functional systems, i.e., ISAC [5]. This convergence creates possibilities for enhanced spectral efficiency, beamforming efficiency, and improved cost and size efficiency in integrated functionalities within 6G networks [7].

ISAC is emerging as a pivotal technology across various application scenarios. For instance, ISAC can aid in autonomous driving by significantly impacting traffic congestion management, enhancing safety measures, and improving overall reliability, primarily through high-resolution obstacle detection [26]. Moreover, the convergence of ISAC with UAVs opens up opportunities for offloading communication and sensing tasks onto ISAC, leading to a mutual enhancement in capacity and flexibility [27]. Notably, there is a growing enthusiasm reflected in recent publications concentrating on various aspects of ISAC, including waveform design [28,29,30], coding design [31,32,33], experimental testing [34,35,36], and signal processing [37].

### 4.1. ISAC Design Philosophy

ISAC aims to integrate sensing and communication and achieve direct trade-offs and mutual benefits. To illustrate these, let us consider a general ISAC system having an *M* ULA antenna base station (BS), *K* single-antenna users, and *N* targets. The BS transmitted signal at the *l*-th time slot, $x\left(l\right)\in C^{M\times 1}$, can be given as:(11)$$\begin{array}{c} x\left(l\right)=\sum_{k\in K}\sqrt{\rho }w_{k}s_{k}\left(l\right)+\sqrt{\left(1-\rho \right)}s_{t}\left(l\right), \end{array}$$
where $s_{k}\left(l\right)$ and $w_{k}\in C^{M\times 1}$ are the intended data and the communication beamforming for the *k*-th user, respectively. In (Equation 11), $s_{t}\left(l\right)∼\mathrm{CN}\left(0,R_{t}\right)$ is the sensing signal with the covariance matrix of $R_{t}=E\left[s_{t}s_{t}^{H}\right]⪰0$, which is designed to extend the degrees of freedom of the BS transmit signal to achieve enhanced sensing performance [15]. Finally, 0≤ρ≤1 is the power allocation factor between communication and sensing, which determines the priority and the level of integration between communication and sensing [15].

Depending on the level of integration, ISAC design falls into three different categories: (i) Communication-centric design with high ρ (ρ→1), (ii) sensing-centric design with low ρ (ρ→0), and (iii) joint design with moderate ρ [38].

*Communication-centric design:* This term pertains to the design of a communication signal that can serve a dual purpose for sensing as well [39]. It uses the minimum amount of modification to incorporate wireless sensing. Let us consider a downlink communication and sensing system. A straightforward approach involves utilizing the communication signal and extracting target information from the echoes.*Sensing-centric design:* Sensing has a higher priority than communication, i.e., using the radar (sensing) for communication as a secondary function [40]. For example, wireless communication capacity can be added to a radar sensor by embedding communication symbols in the output waveform. In practice, the information contained within the sensing signal should not compromise the integrity of the sensing function [40].*Joint design:* Within this classification, the signal is collaboratively designed with equal/well-designed priorities for sensing and communication to achieve improved trade-offs between two functionalities, such as a more flexible resource allocation framework between sensing and communication functions. Consequently, the joint signal design offers greater flexibility and higher DoF to effectively balance the requirements of both sensing and communication [41].

With these diverse design principles, ISAC elevates traditional wireless communication networks to a new dimension and profoundly impacts the current information society [15]. In particular, sensing capabilities might become a standard feature in next-generation wireless networks, serving as both an auxiliary approach and a fundamental service for many users and applications. On the other hand, sensory data can improve communication performance, e.g., sensing-aided vehicle beamforming and resource management. Furthermore, with sensing capabilities, future mobile networks can observe their surroundings and become more intuitive [15]. These networks continuously perceive their surroundings and offer services like monitoring traffic, weather, and human activity recognition. These collected/sensed data are the foundation for developing intelligence for the ISAC network and related applications, including smart homes, transportation, and cities.

### 4.2. ISAC Applications

This subsection overviews some potential application scenarios facilitated by ISAC’s communication and sensing capabilities (Figure 4).

#### 4.2.1. Human Activity Recognition

Activity recognition enables computing systems to monitor, analyze, and support individuals in their daily lives by recording their behaviors. These data have several potential applications, particularly in the healthcare industry [42]. As over-the-air signals are influenced by both static and moving objects, as well as dynamic human activities, amplitude/phase variations in wireless signals could be used to detect or recognize human presence, proximity, falls, sleep, breathing, and daily activities by extracting range, Doppler, or micro-Doppler features. Recognizing or detecting a driver’s blink rate with a high sensing resolution can help detect fatigued driving to enhance road safety. In addition, integrating sensing capability into current commercial wireless equipment, such as Wi-Fi devices, can detect and identify residents’ behaviors, resulting in an innovative and human-centric living environment [42].

#### 4.2.2. Localization and Tracking

Localization is essential for standardizing, implementing, and utilizing all generations of networks. Due to bandwidth and antenna restrictions, resulting in low range and angle resolutions, existing cellular networks (e.g., fourth generation (4G) and 5G) can only give measurement data with meter-level accuracy to aid with global navigation satellite systems [10,43]. According to the 5G New Radio (NR) Release 17 key parameter indications, the highest needed horizontal/vertical localization accuracies in industrial IoT applications are 0.2m/1m. However, the current systems fail to meet this requirement and, hence, cannot facilitate future applications. In particular, indoor human activity identification, autonomous robotics, and manufacturing demand improved location precision to pinpoint user placements. Current wireless localization technologies rely on device-based implementation, in which wireless equipment (e.g., a smartphone) determines the location of an object through signal interactions and geometrical relationships with other Wi-Fi access points or BSs. However, this method restricts the number of objects that may be located and does not apply to diverse scenarios [10,43].

Nevertheless, ISAC-enabled cellular networks can achieve greater localization accuracy than existing localization systems by using extra Doppler processing and essential information from multi-path components. Additionally, a cellular network with sensing capabilities is useful for more than just detecting the position of a specific object using a smartphone; it can also collect spectroscopic and geometric information from the surrounding environment.

#### 4.2.3. V2X

Autonomous vehicles have the potential to revolutionize the transportation business by improving highway capacity and traffic flow, reducing fuel consumption and pollution, and lowering accident rates [44,45]. To accomplish this, automobiles are equipped with communication transceivers and various sensors that extract ambient information while exchanging data with roadside units (RSUs), other vehicles, and even pedestrians [44,45]. ISAC also addresses electromagnetic compatibility and spectrum congestion issues as a feasible solution. For instance, ISAC-aided V2X can provide environmental information for quick vehicle platooning, secure and seamless access, and simultaneous localization and mapping (SLAM) [44,45]. In addition, RSU networks can expand the sensing range of a passing vehicle beyond its own LoS and field of view.

#### 4.2.4. Smart Manufacturing and Industrial IoT

Wireless networks in sectors such as construction, automobile manufacturing, and product lines have sparked the industrial IoT revolution, resulting in orders of magnitude gains in automation and production efficiency [46]. Such cases include network nodes and robots collaborating to complete complicated and often sensitive tasks that need large-scale connection and impose severe latency constraints. Nevertheless, ISAC can provide significant benefits in such smart factory situations, in addition to the ultra-fast, low-latency connections. By integrating sensing functions, industrial nodes and robots can navigate, coordinate, and map their surroundings more efficiently, potentially reducing signaling overhead [46].

#### 4.2.5. Environmental Monitoring

Another compelling application of ISAC is environmental monitoring, including areas such as weather prediction, pollution detection, and precipitation monitoring [47]. Communication signals across different frequencies can be leveraged to monitor environmental changes [10]. For instance, mmWave signals are close to the water vapor absorption bands, making them sensitive to humidity [10]. Analyzing data from mmWave links in conjunction with other factors like path loss makes it possible to monitor atmospheric conditions effectively. This capability allows ISAC systems to provide real-time environmental insights, making them valuable tools for comprehensive environmental monitoring.

#### 4.2.6. NF-ISAC Use Cases

While NF-ISAC and conventional ISAC may share some common use cases, there are also unique use cases with features that are specific to NF-ISAC. NF-ISAC systems can deliver high-precision positioning and navigation by exploiting the fine spatial resolution available in the NF region. This capability is precious in environments where GPS signals are weak or unavailable, such as inside buildings or warehouses [20]. Additionally, NF-ISAC systems can accurately detect and classify human movements using radar and other sensing techniques, making them ideal for monitoring daily activities and health-related behaviors, such as remote patient monitoring and fitness tracking [15].

Furthermore, NF-ISAC systems can be optimized for efficient wireless power transfer (WPT) and energy harvesting (EH) by leveraging the focused electromagnetic fields in the NF region, which is particularly beneficial for powering small IoT devices or sensors in remote or hard-to-reach locations. Moreover, NF-ISAC can eliminate the need for explicit synchronization between reference stations, thereby reducing system overhead [20].

### 4.3. Industry Progress and Standardization

As the groundwork for 6G research gains momentum, ISAC has emerged as a focal point, garnering substantial attention from key industry players. Major companies like Ericsson, NTT DOCOMO, ZTE, China Mobile, China Unicom, Intel, and Huawei have all underscored the pivotal role of sensing in their 6G white papers and Wi-Fi 7 visions, emphasizing its significance in shaping the future of wireless technology [48,49,50]. In particular, ref. [51] define harmonized sensing and communication as one of the three emerging opportunities in 5.5G, i.e., beyond 5G, with the primary goal of using the sensing capabilities of existing massive MIMO BSs and enabling future UAVs and automotive vehicles. In [50], Huawei envisions the 6G new air interface supporting simultaneous wireless communication and sensor signals. This allows ISAC-enabled cellular networks to “see” the actual/physical environment, which is one of 6G’s distinguishing features. In [52], Nokia also introduces a unified mmWave system as a model for future indoor ISAC technologies.

Furthermore, the Institute of Electrical and Electronics Engineers (IEEE) standardization association and the 3rd Generation Partnership Project (3GPP) have substantially contributed to the evolution of ISAC-related specifications. Notably, IEEE 802.11 established the wireless local-area network (WLAN) Sensing Topic Interest Group and Study Group in 2019, followed by the creation of the official Task Group IEEE 802.11bf in 2020. These initiatives aim to define the necessary modifications to existing Wi-Fi standards, aligning them with 802.11-compliant waveforms to enhance sensing capabilities.

In parallel, the NR Release 16 specification has redefined the positional reference signal, endowing it with a more regular signal structure and a significantly wider bandwidth. This redesign facilitates improved signal correlation and parameter estimation, including precise time-of-arrival estimation, demonstrating the concerted efforts across industry and standards bodies to advance ISAC technology [53].

## 5. Near-Field Integrated Sensing and Communications

NF-ISAC must optimize communication and sensing performance on a shared system architecture and hardware platform to meet future network requirements. To accomplish this, the impacts of NF channels and transmission must be considered when balancing communication and sensing operations and their trade-offs for narrow-band and wide-band systems.

### 5.1. Near-Field Narrow-Band Systems

For effective joint communication and sensing, NF beam-focusing designs must alleviate severe path loss in high-frequency bands. Compared to conventional time-division-based beamforming, which alters communication and sensing beams at separate times, it is more promising to generate multiple beams concurrently to enhance communication and sensing performance [2]. In particular, communication beams should be focused on users at specific locations/regions to improve NF communication performance. Conversely, the detection/estimation goal primarily determines NR sensing beam control. Sensing beams should dynamically scan the region of interest in the angular and range domains to estimate target parameters (e.g., angle, range, and Doppler). The multi-beam design based on array partition, which divides the ELAA into many sub-arrays, each responsible for controlling a sub-beam, is one simple yet effective existing approach. However, it limits the number of antennas allocated per sub-array; thus, communication users or sensing targets may be located in the FF of the sub-array, reducing the beam-focusing effect. To strike the communication and sensing performance trade-off, it is required to precisely determine the number of sub-arrays and optimize the antenna allocation for these sub-arrays.

Conversely, target tracking necessitates adaptive beam designs to track moving targets. Both communication and sensing beams must be directed toward communication users and/or sensing targets to boost communication and sensing SNR. Hence, although there are weak channel correlations, multiple beams must be generated to ensure high communication and sensing SNR when the targets and communication users are at the same angle but at different ranges [54]. This contrasts significantly with the FF region, where only one beam is required when they exist at the same angle.

Narrow-band ISAC is suitable for applications requiring low-power, low-latency sensing and communication in NF settings [2,54]. Examples include smart homes, IoT devices, human–machine interaction, automotive applications, and security and surveillance. Narrow-band ISAC, used in smart homes and IoT devices, allows for efficient home automation, wearable technology, and human–machine interaction via proximity detection and gesture recognition. For instance, smart speakers may recognize user motions or identify people nearby while interacting with other devices. It is also useful in automotive applications that need in-cabin sensing, such as monitoring driver alertness or operating infotainment systems by gesture. Furthermore, NF-narrow-band ISAC is critical in security and surveillance as it can provide short-range proximity detection to prevent unauthorized access while maintaining continuous communication with central security systems [2,54].

### 5.2. Near-Field Wide-Band Systems

Wide-band NF-ISAC is more complex due to the prominent beam-split effect. In particular, as the operating frequency affects the NF beam-focusing location/region, beams at different frequencies in wide-band systems concentrate at different locations, affecting communication and sensing performance [55].

For communication users, the beam-split effect degrades received SNR as the beam energy distributes in different places [55]. One feasible option for addressing this issue is integrating true-time-delay (TTD) devices with ELAA to generate frequency-dependent beams. TTDs can introduce adjustable time delays into signals, resulting in frequency-dependent phase changes. Conversely, the NF beam-split effect can efficiently improve sensing performance. Specifically, the beam-split effect can generate numerous beams simultaneously, each focused at the same range but at different angles [56]. There is a fundamental trade-off between TTD allocation for communication and sensing. Furthermore, various design challenges, such as how to improve radar sensing performance (e.g., Cramér-Rao bound (CRB) minimization) under communication performance constraints and vice versa, must be addressed.

Wide-band ISAC is ideal for high-resolution sensing and data-rich communication in NF environments [55,56]. It can be used in imaging radar systems for environmental mapping, detecting objects, and identification, which is especially useful in robotics and medical health monitoring. Wide-band ISAC improves the user experience in augmented and virtual reality systems by allowing high-fidelity interaction, such as precise hand-tracking and object recognition, while maintaining high-speed communication with processors. Automotive applications also benefit from wide-band ISAC, such as advanced driver assistance systems, which use high-precision sensing for parking assistance, blind-spot identification, and collision avoidance. Additionally, wide-band ISAC is essential in high-resolution security systems, such as biometric sensing for fingerprint or facial identification, where detailed sensing must be combined with secure, reliable communication.

## 6. Near-Field ISAC Literature Survey

As mentioned earlier, the shift towards higher frequency bands, combined with the incorporation of extensive antenna elements, in the future of wireless communication (6G), leads to a broadened NF region in contrast to previous generations (1G–5G). This transition to a higher band and the utilization of massive antenna arrays yield high-accuracy and high-resolution sensing capabilities, thereby facilitating the implementation of ISAC. Hence, various previous studies have addressed diverse facets such as beamforming and waveform design, system optimization, and performance analysis within the NF integrated with the ISAC system.

For instance, in the work by Luo et al. [57], researchers investigate the NF user localization by leveraging the beam squint effect. Here, the BS utilizes orthogonal frequency-division multiplexing (OFDM) signals to detect users in the frequency domain, thereby diminishing the overhead associated with beam sweeping. Similarly, the research presented in [58] illustrates that employing OFDM waveforms alongside large arrays in full-duplex (FD) massive MIMO BS facilitates precise localization even within a narrow bandwidth. Examining the self-interference bottleneck within the FD system, ref. [59] delves into the transceiver design of the ISAC system within the THz frequency band, featuring FD-MIMO metasurface antenna panels at the BS. The objective is to devise dynamic metasurface antenna beamforming matrices alongside digital transmit beamforming matrices and digital self-interference cancellation matrices. In a similar vein, ref. [60] tries to minimize the weighted summation of radar and communication beamforming error in a multi-user single-target system. The Cramer–Rao bound (CRB) for target parameter estimate is derived to confirm the effectiveness of NF beamforming. The CRB is formulated and minimized in the study by Wang et al. for NF joint distance and angle sensing [12]. The investigation explores two scenarios: (1) employing fully digital antennas, where each antenna at the BS is linked to a dedicated RF chain, and (2) utilizing hybrid digital and analog antennas, incorporating large-dimensional analog phase shifters alongside low-dimensional digital components. The findings indicate that NF favors ISAC more than FF. In NF multi-target detection, ref. [61] pioneered a beamforming design for an NF-ISAC system with multi-user, multi-target, and FD BS. It presented an iterative beamforming approach to reduce BS transmit power by designing optimal transmit (communication and sensing) and reception beamforming at the BS.

WPT is a significant application of NF communication, facilitating the charging of electronic devices like smartphones, smartwatches, and electric toothbrushes, as well as enabling wireless charging pads and mats. Notably, the distinction between NF WPT and FF WPT lies in the efficiency drop of power transfer as the system transitions from NF to FF conditions [62]. In light of this, ref. [63] delves into the study of transmit beam pattern matching and max-min beam pattern gain in radar sensing and WPT systems. The findings underscore NF’s superior transmit beam pattern gain compared to the FF scenario.

To address the multi-functionality demands of future wireless communication systems, ref. [64] investigates a NOMA-assisted ISAC multi-tier computing (MTC) system. In this proposed framework, a multi-functional BS conducts target sensing while concurrently offering edge computing services to nearby users, with the ability to offload computation tasks to a robust cloud server. Results demonstrate the superiority of the MTC system over single-tier computing systems, with NOMA playing a crucial role in ensuring high-quality computing services.

Furthermore, RIS technology is closely intertwined with 6G applications. While RIS comprises numerous passive reconfigurable elements, its integration with cellular networks can dynamically alter the wireless propagation environment [65,66]. As a result, ISAC can harness the additional spatial degrees of freedom facilitated by integration with RIS. In study [65], the focus lies on RIS-assisted NF localization. The polar-domain gradient descent algorithm and multiple signal classification (MUSIC) demonstrate superior angle accuracy compared to existing algorithms. In [66], the RIS enhances the transmission link by establishing LoS paths between the transmitter and receiver in NF scenarios. The continuous RIS design aims to transform planar waves into cylindrical or spherical waves, thus focusing energy on the ULA or the single antenna receiver. Furthermore, the maximum likelihood method and focal scanning are utilized for receiver localization, with corresponding position error analysis conducted.

## 7. Case Study and Discussion

The NF-ISAC design philosophies were outlined in Section 4.1. Based on them, this section develops several case studies.

In addition to the design philosophies, we consider NOMA to facilitate multiple access for communication users. NOMA has been extensively studied as an efficient multiple-access technique, primarily in FF scenarios. However, when considering NF-NOMA, as opposed to its well-established use in FF, the transition is not merely about repurposing existing FF concepts. Instead, NF presents unique opportunities and challenges that can lead to innovations and adaptations for NOMA. In particular, NOMA could develop in the NF in the following aspects:*Exploiting spatial resolution:* Unlike conventional FF, NF spherical wavefronts enable finer spatial resolution. Thus, NOMA can use this to effectively separate users in the spatial domain, resulting in enhanced user differentiation even when users are closely located. Additionally, the NF environment enables more precise user clustering based on spatial features such as distance, angle, and position. NOMA can be designed to exploit these features, optimizing power allocation and interference management more effectively than in the FF.*Advanced beamforming:* NF beamforming is much more precise due to the ability to focus beams on a specific point in space rather than just a direction. NOMA can leverage this by enabling multiple users to share the same spatial resources with carefully designed power levels, leading to higher spectral efficiency.*Enhanced interference management:* NF beam-focusing improves interference management. NOMA can exploit this to develop new interference cancellation strategies that utilize the precise spatial information offered by NF, minimizing inter-user interference more effectively than FF. Furthermore, NF NOMA can employ power control techniques that include the user’s distance, spatial location, and angle, resulting in more efficient interference mitigation and resource utilization.

While some concepts from FF NOMA can be repurposed for NF, NF presents unique opportunities that require NOMA to evolve and adapt. The NF’s ability to utilize fine spatial resolution, beam focusing, and wavefront modification offers new avenues for NOMA in user distinction, beamforming, and interference management. Thus, NOMA in NF is more than just repeating FF notions; it is about utilizing NF’s distinctive characteristics to establish more intelligent, efficient, and secure communication systems.

### 7.1. Preliminaries

#### 7.1.1. System and Channel Models

Figure 5 investigates a downlink (DL) NOMA ISAC network. It comprises an *M* ULA antenna BS, *K* single-antenna NOMA users, and a sensing target. Without loss of generality, the ULA antennas are at a spacing of *d*, and the coordinate system’s origin is at the center of the ULA.

*NF-channel model:* The NF spherical wave channel model for a ULA in Section 3.1 is adopted here. The coordinates of the *m*-th element of ULA, the *k*-th user, and the target are $s_{m}={\left[md,0\right]}^{T}$ for $m\in \{-\hat{M},\ldots ,\hat{M}\}$, $u_{k}={\left[r_{k}\cos\left(\theta_{k}\right),r_{k}\sin\left(\theta_{k}\right)\right]}^{T}$ for k∈K, and $u_{t}={\left[r_{t}\cos\left(\theta_{t}\right),r_{t}\sin\left(\theta_{t}\right)\right]}^{T}$, respectively. In addition, $\theta_{k}$ and $\theta_{t}$ are the azimuth angle of the *k*-th user and the target, respectively, and  $r_{k}=∥u_{k}-s_{0}∥$ denotes the distance between *k*-th user and center of ULA BS antenna. Using (Equation 6) and (Equation 5), the NF-LoS channel between ULA BS and *k*-th user can be written as [2]:(12)$$\begin{array}{c} h_{k,\mathrm{NF}}^{\mathrm{LoS}}=\beta e^{-j\frac{2\pi }{\lambda }r_{k}}\left[e^{-j\frac{2\pi }{\lambda }\left(\hat{M}d\cos\left(\theta_{k}\right)+\frac{{\hat{M}}^{2}d^{2}\sin{\left(\theta_{k}\right)}^{2}}{2r_{k}}\right)},\ldots ,e^{-j\frac{2\pi }{\lambda }\left(-\hat{M}d\cos\left(\theta_{k}\right)+\frac{{\hat{M}}^{2}d^{2}\sin{\left(\theta_{k}\right)}^{2}}{2r_{k}}\right)}\right], \end{array}$$
where β is the channel gain, which is assumed to have an equal value for all antenna elements [2]. Assuming all users and the target are in the ULA’s NF region, only one line-of-sight (LoS) channel between the users/target and the BS is considered. Nonetheless, our framework is adaptable to multi-path and non-LoS scenarios.

While this study primarily focuses on NF channels due to the extended range of NF–FF boundaries at higher frequencies, the FF channel model is also employed as a comparative benchmark.

*FF-channel model:* The distance between the BS and the users/target for the FF becomes r≫D. Therefore, the NF channel model in (Equation 12) reduces into following model:(13)$$\begin{array}{cc} h_{k,\mathrm{FF}}^{\mathrm{LoS}} & =\beta e^{-j\frac{2\pi }{\lambda }r_{k}}\left[e^{-j\frac{2\pi }{\lambda }\hat{M}d\cos\left(\theta_{k}\right)},\ldots ,e^{j\frac{2\pi }{\lambda }\hat{M}d\cos\left(\theta_{k}\right)}\right]. \end{array}$$

In the sequel, the channel between the *k*-th user and the BS is $h_{k}\in C^{M\times 1}$, and the channel between the target and the BS is $g_{t}\in C^{M\times 1}$.

#### 7.1.2. Transmit Signal Model

The BS transmits the information-bearing signal $s_{k}∼\mathrm{CN}\left(0,1\right)$ for the *k*-th user (k∈K), along with the dedicated sensing signal $s_{t}∼\mathrm{CN}\left(0,R_{t}\right)$. Here, $R_{t}=E\left[s_{t}s_{t}^{H}\right]⪰0$ is the covariance matrix of the sensing signal, which is designed to extend the degrees-of-freedom of the BS transmit signal to achieve enhanced sensing performance [67]. Signals $\left\{{\left\{s_{k}\right\}}_{k\in K},s_{t}\right\}$ are statistically independent from each other [67]. Thus, the BS broadcast signal may be expressed as:(14)$$\begin{array}{c} x=\sum_{k\in K}w_{k}s_{k}+s_{t}, \end{array}$$
where $w_{k}$ denotes the transmit beamforming vector assigned for signal $s_{k}$.

Each user receives the BS signal x and must extract the information sent for itself. In this process, the sensing and other user signals act as interference. Consequently, the received signal at the *k*-th user can be written as:(15)$$\begin{array}{c} y_{k}=h_{k}^{H}w_{k}s_{k}+\underset{\text{inter-user}\;\mathrm{interference}}{\underset{︸}{\sum_{i\in K_{k}}h_{k}^{H}w_{i}s_{i}}}+\underset{\mathrm{sensing}\;\mathrm{interference}}{\underset{︸}{h_{k}^{H}s_{t}}}+n_{k}, \end{array}$$
where $n_{k}∼\mathrm{CN}\left(0,\sigma_{k}^{2}\right)$ is the additive white Gaussian noise (AWGN) at the *k*-th user with zero mean and $\sigma_{k}^{2}$ variance.

Conversely, the BS uses the received echo signal from the target for sensing. The received echo signal at the BS from the target is thus expressed as:(16)$$\begin{array}{c} y_{t}=G_{t}x+n_{t}, \end{array}$$
where $G_{t}≜g_{t}g_{t}^{H}\in C^{M\times M}$ due to the reciprocity in the time division duplex mode [68,69], and $n_{t}∼\mathrm{CN}\left(0,\sigma_{t}^{2}I_{M}\right)$ is the AWGN at the BS. The BS then applies the receiver beamforming $u\in C^{M\times 1}$ to the received signal (Equation 16) to capture the sensing information from the target. The post-processed signal for the target’s sensing information can be given as:(17)$$\begin{array}{c} y_{t}=u^{H}y_{t}=u^{H}G_{t}x+u^{H}n_{t}. \end{array}$$

### 7.2. Communication and Sensing Performance

The sensing and communication rates/signal-to-noise-to-interference ratios (SINRs) significantly affect the performance of both systems. Here, the users’ and the target’s sensing rates are used to analyze and optimize the NF-ISAC system.

#### 7.2.1. Communication Performance

From Equation (Equation 15), the received SINR at the *k*-th user is given as:(18)$$\begin{array}{c} \gamma_{k}=\frac{|h_{k}^{H}w_{k}{|}^{2}}{\sum_{i\in K_{k}}{\left|h_{k}^{H}w_{i}\right|}^{2}+h_{k}^{H}R_{t}h_{k}+\sigma_{k}^{2}}. \end{array}$$
Thereby, the rate of the *k*-th user can be approximated as:(19)$$\begin{array}{c} R_{k}=\log_{2}\left(1+\gamma_{k}\right). \end{array}$$

#### 7.2.2. Sensing Performance

The BS learns and obtains environmental information using the post-processed reflected signal from the target in Equation (Equation 17). Thus, the sensing SINR of the target at the BS is given by:(20)$$\begin{array}{c} \gamma_{t}=\frac{E\left\{|u^{H}G_{t}{x|}^{2}\right\}}{E\left\{|u^{H}n_{t}{|}^{2}\right\}}=\frac{u^{H}G_{t}R_{x}G_{t}^{H}u}{u^{H}\sigma_{t}^{2}I_{M}u}, \end{array}$$
where $R_{x}≜E\left\{xx^{H}\right\}=\sum_{k\in K}w_{k}w_{k}^{H}+R_{t}$ is the covariance matrix of the BS transmitted signal [67].

**Remark** **1.**
*The associated SINRs primarily determine the communication and sensing performance of ISAC systems. In particular, the probability of detecting a communication symbol increases monotonically with SINR [70,71]. Maximizing SINR eventually minimizes the symbol error probability. Hence, the communication SINR (or rate) is a standard performance metric. Similarly, in sensing, the detection probability of a target is proportional to its sensing SINR [67,72]. The sensing SINR enables target detection using both transmit and receiver beamforming. It also aids in reducing interference between targets. However, the standard mean squared error of the transmit beam pattern does not account for the receiver beam pattern or target interference. Given the benefits of sensing SINR, it is a popular metric for sensing performance.*


As mentioned in Section 4.1, the NF-ISAC systems can also be classified into three categories based on the design target. This section examines those and defines an optimization problem tailored to the criteria of each class.

### 7.3. Case 1: Sensing Performance Maximization

In Case 1, the priority is maximizing sensing performance over communication. Subsequently, the optimization problem can be expressed as:
(21a)$$\begin{array}{cc} \left(P1\right):\max_{\left\{w_{k}\right\},R_{t},u}\; & \gamma_{t}\left(\left\{w_{k}\right\},R_{t},u\right), \end{array}$$
(21b)$$\begin{array}{cc} s.t.\; & \gamma_{k}\left(\left\{w_{k}\right\},R_{t}\right)\geq \gamma_{th},\;\;\forall k\in K, \end{array}$$
(21c)$$\begin{array}{cc} \; & \sum_{k\in K}{∥w_{k}∥}^{2}+\mathrm{Tr}\left(R_{t}\right)\leq P_{b}, \end{array}$$
(21d)$$\begin{array}{cc} \; & R_{t}⪰0, \end{array}$$
where $\gamma_{th}$ is the minimum SINR threshold required by the *k*-th user and $P_{b}$ is the maximum allowable transmit power of the BS. In (P1), ([Disp-formula FD21b-entropy-26-00773]) is to satisfy the minimum SINR required for each user, ([Disp-formula FD21c-entropy-26-00773]) is the total transmit power constraint at the BS, and  ([Disp-formula FD21d-entropy-26-00773]) is the inherent constraint of the sensing signal. Problem (P1) is non-convex because of its objective and constraints, including the optimization variables’ products. To address this, alternating optimization (AO) is utilized [73].

#### Proposed Solution

To solve problem (P1), it is decomposed into two distinct subproblems involving two-variable blocks using AO techniques: $w_{k},R_{t}$ and u. Specifically, the subsequent subsection focuses on maximizing sensing SINR by jointly optimizing the design of $w_{k}$ and $R_{t}$ while keeping u constant. Next, u is optimized while maintaining the fixed values of $w_{k},R_{t}$. This procedure repeats until convergence or the stopping criterion is satisfied. This strategy is useful when direct or simultaneous optimization of all variables is difficult or computationally costly [73]. The details are as follows:

*Optimizing $\left\{w_{k}\right\}$ and $R_{t}$:* To solve this subproblem, the semidefinite relaxation (SDR) technique can be applied. Denote matrix $W_{k}=w_{k}w_{k}^{H}$, where $W_{k}$ is semidefinite and satisfies rank-one constraint. Consequently, problem (P1) can be written as:
(22a)$$\begin{array}{cc} \left(P1.1\right):\max_{\left\{W_{k}\right\},R_{t}}\; & \frac{u^{H}G_{t}\left(\sum_{k\in K}W_{k}+R_{t}\right)G_{t}^{H}u}{u^{H}\sigma_{t}^{2}I_{M}u}, \end{array}$$
(22b)$$\begin{array}{cc} s.t.\; & \frac{\mathrm{Tr}(h_{k}h_{k}^{H}W_{k})}{\sum_{i\in K_{k}}\mathrm{Tr}\left(h_{k}h_{k}^{H}W_{i}\right)+h_{k}^{H}R_{t}h_{k}+\sigma_{k}^{2}}\geq \gamma_{th},\;\;\forall k\in K, \end{array}$$
(22c)$$\begin{array}{cc} \; & \sum_{k\in K}\mathrm{Tr}\left(W_{k}\right)+\mathrm{Tr}\left(R_{t}\right)\leq P_{b}, \end{array}$$
(22d)$$\begin{array}{cc} \; & R_{t}⪰0,\left\{W_{k}\right\}⪰0,\;\;\forall k\in K, \end{array}$$
(22e)$$\begin{array}{cc} \; & \mathrm{rank}(W_{k})=1,\;\;\forall k\in K. \end{array}$$

Note that P1.1 is still non-convex due to the constraint of Equation (22e). Thus, relaxing this rank-one constraint, (P1.1) is transformed into a convex optimization problem, i.e., a conventional semi-definite programming (SDP) problem, and can be solved via Matlab CVX [73]. To optimize $\left\{w_{k}\right\}$, the Gaussian randomization (GR) can be utilized to enforce the desired implicit rank-one constraint on the matrix solution. In particular, let (P1.1)’s optimal solution be $W_{k}^{*}$, which is not necessarily rank-one. If it is, the sole eigenvalue is the trace of $W_{k}^{*}$, and the related eigenvector defines the optimal beamforming vector. Otherwise, a rank-one solution is extracted from $W_{k}^{*}$. To do this, a random vector r can be selected from $\mathrm{CN}(0,W_{k}^{*})$ and projected to the feasible region of P1.1. This random sampling is repeated numerous times, and only the best approximate solution is selected. This method is referred to as GR [74].

*Optimizing u:* It can be noted that only $\gamma_{t}$ is dependent on u. Therefore, for given $\left\{\left\{w_{k}\right\},R_{t}\right\}$, P1 can be expressed as:(23)$$\begin{array}{c} \left(P1.2\right):\max_{u}\;\frac{u^{H}G_{t}R_{x}G_{t}^{H}u}{u^{H}\sigma_{t}^{2}I_{M}u}. \end{array}$$

This is a standard Rayleigh quotient problem [75]. Therefore, the optimal $u^{*}$ is the eigenvector corresponding to the maximum eigenvalue of matrix $B^{-1}A$, where $B=\sigma_{t}^{2}I_{M}$ and $A=G_{t}R_{x}G_{t}^{H}$. The implementation steps are detailed in Algorithm 1.
**Algorithm 1** Algorithm for solving Case 1.1:**Input**: Set the iteration counter x=0, the convergence tolerance ϵ>0, initial feasible solution $\left\{W_{k}^{\left(0\right)}\right\},R_{t}^{\left(0\right)}$. Initialize the objective function value $\gamma_{t}^{\left(0\right)}\left(Z\right)=0$.2:**while** $\frac{\gamma_{t}^{\left(x+1\right)}\left(Z\right)-\gamma_{t}^{\left(x\right)}\left(Z\right)}{\gamma_{t}^{\left(x+1\right)}\left(Z\right)}\geq \epsilon$ **do**3:   Solve (P1.1) and obtain the optimal solution for $\left\{w_{k}^{\left(x+1\right)}\right\}$ by recovering a rank-one solution via GR in Algorithm 2.4:   Solve Equation (Equation 23) and obtain the optimal solution as $u^{\left(x+1\right)}$.5:   Calculate the objective function value $\gamma_{t}^{\left(x+1\right)}\left(Z\right)$.6:   Set x←x+1;7:**end while**8:**Output**: $Z^{*}=\left\{\left\{w_{k}^{*}\right\},R_{t}^{*},u^{*}\right\}$.

**Algorithm 2** Gaussian Randomization.
1:**Input**: matrix F, number of randomization sample *N*, and initial objective value $O^{0}=-\mathrm{Inf}$.2:**Do** Apply eigenvalue decomposition to obtain $F_{op}=U\Lambda U^{H}$.3:**for** n≤N **do**4:   Generate *N* independent unit-variance zero-mean circularly symmetric complex Gaussian vector z.5:   Set $f_{op}=U\Lambda^{0.5}z$ and calculate $O^{n+1}\left(f_{op}\right)$.6:   **if** $O^{n+1}\geq O^{n}$ **then**7:     $O^{n}\leftarrow O^{n+1}$, $f_{op}=U\Lambda^{0.5}z$8:   **end if**9:
**end for**
10:**Output**: $f_{op}$.


### 7.4. Case 2: Communication Performance Maximization

If communication functionality is of higher priority, the design strategies may delve into maximizing communication performance. The communication-centric optimization problem can be expressed as follows:
(24a)$$\begin{array}{cc} \left(P2\right):\max_{\left\{w_{k}\right\},R_{t},u}\; & \sum_{k\in K}R_{k}\left(\left\{w_{k}\right\},R_{t}\right), \end{array}$$
(24b)$$\begin{array}{cc} s.t.\; & \gamma_{t}\left(\left\{w_{k}\right\},R_{t},u\right)\geq \eta , \end{array}$$
(24c)$$\begin{array}{cc} \; & \sum_{k\in K}{∥w_{k}∥}^{2}+\mathrm{Tr}\left(R_{t}\right)\leq P_{b}, \end{array}$$
(24d)$$\begin{array}{cc} \; & R_{t}⪰0, \end{array}$$
where η is the minimum required sensing SINR. Adopting a method like the one described in Section 7.3 for optimizing the combiner vector u, the optimal combiner vector for this problem can be obtained by solving (Equation 23). Consequently, to maintain conciseness, specific details are excluded from this discussion.

Therefore, the optimization problem over beamformers and sensing covariance matrix can be written as:
(25a)$$\begin{array}{cc} \left(P2.1\right):\max_{\left\{w_{k}\right\},R_{t}}\; & \sum_{k\in K}R_{k}\left(\left\{w_{k}\right\},R_{t}\right), \end{array}$$
(25b)$$\begin{array}{cc} s.t.\; & (24b),(24c),(24d). \end{array}$$
The objective of (P2.1) is non-convex due to the fractional form terms and quadratic form of $w_{k}$. To address this issue and transform the problem into a convex formulation, use the SDR with matrix definition $W_{k}=w_{k}w_{k}^{H}$ satisfying rank-one constraint. Then, fractional programming (FP) [76] is employed. To invoke FP techniques, one utilizes an auxiliary variable and the Lagrangian dual transform $\gamma ={\left[\gamma_{1},\ldots ,\gamma_{K}\right]}^{T}$. Therefore, the objective in (P2.1) can be rewritten as:(26)$$\begin{array}{cc} {\bar{R}}_{\mathrm{sum}}\left(\gamma ,\left\{W_{k}\right\},R_{t}\right)= & \sum_{k\in K}\log_{2}\left[1+2\gamma_{k}\sqrt{h_{k}^{H}W_{k}h_{k}}-\right.\\ \; & \left.\gamma_{k}^{2}\left(\sum_{i\in K_{k}}h_{k}^{H}W_{i}h_{k}+h_{k}^{H}R_{t}h_{k}+\sigma_{k}^{2}\right)\right]. \end{array}$$

Subsequently, problem (P2.1) can be rewritten as:
(27a)$$\begin{array}{cc} \left(P2.2\right):\max_{\left\{W_{k}\right\},R_{t},\gamma }\; & {\bar{R}}_{\mathrm{sum}}, \end{array}$$
(27b)$$\begin{array}{cc} s.t.\; & \frac{u^{H}G_{t}\left(W_{k}+R_{t}\right)G_{t}^{H}u}{u^{H}\sigma_{t}^{2}I_{M}u}\geq \gamma_{th},\;\;\forall k\in K, \end{array}$$
(27c)$$\begin{array}{cc} \; & \sum_{k\in K}\mathrm{Tr}\left(W_{k}\right)+\mathrm{Tr}\left(R_{t}\right)\leq P_{b}, \end{array}$$
(27d)$$\begin{array}{cc} \; & W_{k}⪰0,\;\;\forall k,\;R_{t}⪰0, \end{array}$$
(27e)$$\begin{array}{cc} \; & \mathrm{rank}(W_{k})=1,\;\;\forall k. \end{array}$$

Problem (P2.2) remains non-convex due to the rank-one constraint. To resolve this, the constraint is dropped, resulting in a convex optimization problem for $\{W_{k}\},$ and $R_{t}$, which can be solved using Matlab CVX [73]. However, the AO algorithm must be deployed to update the auxiliary variable γ and optimize $\{W_{k}\},$ and $R_{t}$ iteratively. The optimal γ in a *x* iteration of AO algorithm can be determined by solving $\frac{\partial {\bar{R}}_{\mathrm{sum}}}{\partial \gamma_{k}}=0$ and which yields
(28)$$\begin{array}{cc} \gamma_{k} & =\frac{\sqrt{h_{k}^{H}W_{k}^{\left(x\right)}h_{k}}}{\sum_{i\in K_{k}}h_{k}^{H}W_{i}^{\left(x\right)}h_{k}+h_{k}^{H}R_{t}h_{k}+\sigma_{k}^{2}}. \end{array}$$

Afterward, the Gaussian randomization process [77] can be deployed to ensure the implicit satisfaction of the rank-one constraint. Algorithm 3 presents the steps for solving (P2).
**Algorithm 3** Algorithm for solving Case 2.1:**Input**: Set the iteration counter x=0, the convergence tolerance ϵ>0, initial feasible solution $\left\{W_{k}^{\left(0\right)}\right\},R_{t}^{\left(0\right)},\gamma^{\left(0\right)}$. Initialize the objective function value ${\bar{R}}_{\mathrm{sum}}^{\left(0\right)}\left(Z\right)=0$.2:**while** $\frac{{\bar{R}}_{\mathrm{sum}}^{\left(x+1\right)}\left(Z\right)-{\bar{R}}_{\mathrm{sum}}^{\left(x\right)}\left(Z\right)}{{\bar{R}}_{\mathrm{sum}}^{\left(x+1\right)}\left(Z\right)}\geq \epsilon$ **do**3:   **Do:** Solve (P2.2) and obtain solution as $\left\{W_{k}^{\left(x+1\right)}\right\},R_{t}^{\left(x+1\right)}$.4:   **Update:** $\gamma^{\left(x+1\right)}$ using Equation (Equation 28).5:   **Until:** There is no increment in the objective.6:   Solve Equation (Equation 23) and obtain the optimal solution as $u^{\left(x+1\right)}$.7:   Calculate the objective function value ${\bar{R}}_{\mathrm{sum}}^{\left(x+1\right)}\left(Z\right)$.8:   Obtain $\left\{w_{k}^{\left(x+1\right)}\right\}$ by recovering a rank-one solution via GR and update $\left\{W_{k}^{\left(x+1\right)}\right\}$.9:   Set x←x+1;10:**end while**11:**Output**: $Z^{*}=\left\{\left\{w_{k}^{*}\right\},R_{t}^{*},u^{*}\right\}$.

### 7.5. Case 3: Joint Sensing and Communication Performance Maximization

In this class, sensing and communication priorities can be adjusted. Therefore, the design can be tailored to favor performance trade-offs. Consequently, the optimization issue can be stated as a weighted sum user rate and target as follows:
(29a)$$\begin{array}{cc} \left(P3\right):\max_{\left\{w_{k}\right\},R_{t},u}\; & \rho \sum_{k\in K}R_{k}\left(\left\{w_{k}\right\},R_{t}\right)+\left(1-\rho \right)\log_{2}\left(1+\gamma_{t}\right), \end{array}$$
(29b)$$\begin{array}{cc} s.t.\; & \sum_{k\in K}{∥w_{k}∥}^{2}+\mathrm{Tr}\left(R_{t}\right)\leq P_{b}, \end{array}$$
(29c)$$\begin{array}{cc} \; & R_{t}⪰0. \end{array}$$

Here, 0≤ρ≤1 represents the constant weight that determines the priority of the sum user rate, while 1−ρ indicates the priority of the target sensing rate. By following the analogous procedure to previous problems, it becomes evident that since $\gamma_{t}$ solely relies on the combiner vector u and logarithm in a monotonically increasing function of its argument, the optimal $u^{*}$ that maximizes the rate can be derived by solving Equation (Equation 23). Consequently, this variable is excluded from (P3). Subsequently, similar to case 2, to reformulate this problem in a tractable form, the sum user rate transformation outlined in Equation (Equation 26) is employed. This transformation is then substituted into the sum user rate expression in Equation ([Disp-formula FD29a-entropy-26-00773]) and SDR is used. The final result is as follows:
(30a)$$\begin{array}{cc} \left(P3.1\right):\max_{\left\{W_{k}\right\},R_{t}}\; & \rho {\bar{R}}_{\mathrm{sum}}+\left(1-\rho \right){\bar{R}}_{t}, \end{array}$$
(30b)$$\begin{array}{cc} s.t.\; & \sum_{k\in K}\mathrm{Tr}\left(W_{k}\right)+\mathrm{Tr}\left(R_{t}\right)\leq P_{b}, \end{array}$$
(30c)$$\begin{array}{cc} \; & W_{k}⪰0,\;\;\forall k,\;R_{t}⪰0, \end{array}$$
(30d)$$\begin{array}{cc} \; & \mathrm{rank}(W_{k})=1,\;\;\forall k, \end{array}$$
where
(31)$$\begin{array}{c} {\bar{R}}_{t}=\log_{2}\left(1+\frac{u^{H}G_{t}\left(W_{k}+R_{t}\right)G_{t}^{H}u}{u^{H}\sigma_{t}^{2}I_{M}u}\right). \end{array}$$
However, (P3.1) exhibits non-convexity, attributed to the presence of a non-convex constraint in Equation ([Disp-formula FD30d-entropy-26-00773]). Consequently, dropping this constraint makes (P3.1) a convex problem that can be solved via CVX. To ensure a rank-one solution, Gaussian randomization is deployed. The steps for solving (P3) are outlined in Algorithm 4.
**Algorithm 4** Algorithm for solving Case 3.1:**Input**: Set the iteration counter x=0, the convergence tolerance ϵ>0, initial feasible solution $\left\{W_{k}^{\left(0\right)}\right\},R_{t}^{\left(0\right)},\gamma^{\left(0\right)},\rho^{\left(0\right)}$. Initialize $C_{0}$, $C_{1}$, λ and the objective function value $\chi^{\left(0\right)}\left(Z\right)=0$.2:**while** $\frac{\chi_{\mathrm{sum}}^{\left(x+1\right)}\left(Z\right)-\chi_{\mathrm{sum}}^{\left(x\right)}\left(Z\right)}{\chi_{\mathrm{sum}}^{\left(x+1\right)}\left(Z\right)}\geq \epsilon$ **do**3:   **Do:** Solve (P3.2) and obtain solution as $\left\{W_{k}^{\left(x+1\right)}\right\},R_{t}^{\left(x+1\right)}$.4:   **Update:** $\gamma^{\left(x+1\right)}$ using Equation (Equation 28).5:   **Until:** There is no increment in the objective.6:   Solve Equation (Equation 23) and obtain the optimal solution as $u^{\left(x+1\right)}$.7:   Calculate the objective function value $\chi_{\mathrm{sum}}^{\left(x+1\right)}\left(Z\right)$.8:   Obtain $\left\{w_{k}^{\left(x+1\right)}\right\}$ by recovering a rank-one solution via GR and update $\left\{W_{k}^{\left(x+1\right)}\right\}$.9:   Set x←x+1;10:**end while**11:**Output**: $Z^{*}=\left\{\left\{w_{k}^{*}\right\},R_{t}^{*},u^{*}\right\}$.

### 7.6. Simulation Results and Discussion

This section confirms the effectiveness of the proposed designs by analyzing the simulation results for three specific case studies. Unless otherwise specified, the simulation setup is as follows: M=129 BS antennas [12], uniformly distributed in lines at half-wavelength spacing; the carrier frequency is set to 28 GHz [12]; and there are K=4 [12] users assumed to be randomly located within the Rayleigh distance from the BS. Additionally, a single target is positioned within the Rayleigh distance from the BS and at an angle of −60°. The sensing threshold is η=−10 dB, the user’s QoS threshold is $\gamma_{th}=10$ dB, and the noise power is $\sigma^{2}=-80$ dBm.

#### 7.6.1. Sensing Performance Maximization (Case 1)

Figure 6 plots the achieved sensing SNR as a function of the transmit power at the BS. To comprehensively compare the performance of the proposed system design, the following three baseline schemes are considered. First, the FF performance is provided along with the NF performance for better insight. In the “communication only” scheme, the dedicated signal for sensing is eliminated, allowing the communication signal to perform target sensing. In the “sensing only” scheme, the BS performs sensing exclusively by sending a sensing signal and listening to its echo without serving DL users.

This figure indicates that the NF-proposed scheme performs better than the FF counterpart by approximately 41%. Indeed, this is owing to the NF beam-focusing effect with ELAA at BS and a high operating frequency, which results in a narrow and concentrated beam. However, there is no noticeable difference between the proposed scheme and “communication only” for both NF and FF, as most of the power is assigned to a communication system that can also support the target sensing. Conversely, in “sensing only”, as BS only performs target sensing tasks, BS adjusts its single beam toward the target. Therefore, in most regimes, specifically higher transmit power, better performance is achieved than the proposed scheme, in which BS has dual functionalities, i.e., communication and sensing.

Figure 7 illustrates the achieved beam pattern gain for case study 1, utilizing both NF and FF channel models. In both scenarios, the beams are effectively aligned towards the target angle, aligning to maximize the target’s SNR. However, compared to the FF counterpart, the NF configuration demonstrates a higher beam pattern gain with a narrow beam, i.e., the NF configuration yields a pencil beam, resulting in a narrower and more focused beam due to the beam-focusing effect in NF propagation. Consequently, this enhanced beam gain in the NF scenario can effectively meet the QoS requirements for the user’s SINR.

#### 7.6.2. Communication Performance Maximization (Case 2)

Figure 8 illustrates the achieved sum rate for users by solving Algorithm 3. For comparison, this figure also displays the performance of the FF and communication-only schemes, where no dedicated signal is used for target sensing in the latter. As demonstrated, the NF proposed scheme outperforms the FF scheme by 21 regarding the users’ achieved sum rate. This highlights the advantage of focused and narrow beams in NF, effectively reducing user interference. Conversely, the NF communication-only scheme performs slightly better or the same as proposed. However, as shown in the figure, the FF proposed scheme performs marginally better than its communication-only counterpart.

Figure 9 illustrates the beam pattern gain for Case 2 in a two-user scenario. The beam pattern aligns with the users’ angles to maximize the total user rate. However, the NF gain outperforms the FF gain. Consistent with the findings in Figure 8, the communication-only scheme exhibits lower beam gain because removing the sensing signal causes power to spread toward the users’ beamformers, potentially increasing inter-user interference.

#### 7.6.3. Joint Sensing and Performance Optimization (Case 3)

Figure 10 shows the achieved sum user and sensing rate when both sensing and communication have the same priority, i.e., ρ=0.5. As validated in the previous cases, NF performs much better than the FF, by a magnitude of 52% on average. In the high transmit power regime, the communication-only scheme works better, to an approximate degree of 13% in the NF scheme and 12% in FF.

Figure 11 illustrates the achieved beam pattern gain for case study 3. The beam is clearly aligned towards both the target and user angles as the objective in this case prioritizes both sensing and communication. Comparatively, the communication-only scheme exhibits slightly better beam gain for both NF and FF. Nonetheless, as observed in the previous case, NF outperforms FF regarding beam gain and alignment toward the users and target.

## 8. Future Research Directions

This section addresses open issues, trends, and opportunities for widespread implementation of future NF-ISAC networks.

### 8.1. NF-FF Distance Improvement

The boundary between the NF and FF is not strictly confined. The current phase error-based (i.e., Rayleigh distance) and channel gain error-based approaches fail to encompass all performance metrics comprehensively [78]. On the other hand, when the transmission distance is less than the threshold distance, i.e., the NF–FF boundary, the NF distance is a valuable reference and can be a deciding factor in using NF intelligent spectrum access and control approaches. Consequently, establishing an adequate NF–FF border that considers channel characteristics, signal propagation properties, and various performance criteria is essential to enhancing the overall effectiveness of NF-ISAC.

### 8.2. Accurate NF Channel Models

The NF channel models must be more precise in representing the actual NF EM phenomena [79]. Some existing models frequently treat the NF as equal to the FF counterpart, only accounting for spherical propagation curvature, ignoring the range-dependent amplitude of the received signal, and focusing entirely on phase considerations. Furthermore, they overlook critical NF signal source characteristics such as transmit antenna type, size, and orientation. Consequently, these oversights might considerably influence the signal received by the array [79]. It is thus critical to determine if existing mathematical models effectively describe authentic NF features and to investigate innovative techniques that better capture the intricacies of real-world NF EM behaviors. This could lead to more accurate and reliable results in practical applications.

### 8.3. Channel Estimation

Accurate channel estimation benefits communication and sensing by enabling advanced signal processing, improving spectrum efficiency, increasing localization accuracy, and enhancing reliability [80,81]. In contrast to FF, NF channel estimation offers improved angular resolution and higher channel diversity, allowing more precise and robust estimates. Also, the impacts of multi-path propagation are less prominent than those of FF. This reduced multi-path interference simplifies channel estimation, resulting in more accurate and reliable estimations. However, the structural changes in EM waves caused by ELAAs render conventional channel estimation methods inadequate for NF, prompting the development of revised and unique channel estimation strategies [82,83].

### 8.4. Signal Processing and Low-Complexity Beam-Focusing Designs

Direction-of-arrival (DoA) estimation and beam-focusing (beamforming) are two of the most critical fundamental ISAC signal-processing tasks. DoA estimation determines the angles at which signals arrive at the antenna array. In contrast, beamforming focuses the transmitted and received signals to a specified location while suppressing interference from other directions. Additionally, precise DoA estimation benefits from accurate beamforming and vice versa. With many antenna elements, the scale of beamforming and NF-ISAC optimization challenges becomes significantly large. Standard FF DoA estimation and beamforming designs cannot be directly adapted to NF because they do not account for complicated spatial variations, such as varying signal strengths and phase shifts across the antenna array. Furthermore, FF beams become divergent and wider at close ranges, resulting in higher user interference and angle estimation errors.

### 8.5. Multiple Access

An ELAA introduces an additional spatial resolution based on distance and angle. This and the NF beam-focusing effect can improve the spatial resolution of traditional space-division multiple access (SDMA). Thus, the spatial selectivity of SDMA becomes even more prominent in the NF, effectively exploiting spatial resources for simultaneous data transmission to multiple users [84]. However, it also poses challenges in beam steering and interference management that must be addressed by specifically designed multiple access techniques, such as location division multiple access (LDMA) [84].

### 8.6. ML for NF-ISAC

Machine learning (ML) is a branch of artificial intelligence (AI) that focuses on developing algorithms and statistical models that enable computers to learn and make predictions or decisions based on data. Instead of being explicitly programmed to perform a task, ML algorithms improve their performance through experience and by identifying patterns in data. ML techniques have recently been used in channel estimation and other communication tasks [80,85,86]. ML-based techniques are capable of handling a variety of challenges with NF-ISAC systems. For example, conventional hybrid/digital beamforming techniques, such as zero-forcing with ELAAs, result in excessively high signal processing complexity due to the channel inversion operation of large channel matrices [80]. One solution is to develop tailored deep neural networks that learn effective beamforming using unsupervised deep learning and an appropriate loss function. On the other hand, for NF sensing, ML algorithms can be used to develop NF predictive beam tracking using only a few echo signals [13]. This can significantly increase beam tracking efficiency and reliability while lowering design complexity. Furthermore, ML approaches can be used for NF gesture recognition and motion prediction applications, which rely on NF-received signal features for estimation/recognition.

### 8.7. Integration of NF-ISAC with Other Technologies

Integrating NF-ISAC with other technologies, such as cell-free (CF) architecture, backscatter communication (BackCom), RISs, and others, is essential for unlocking its full potential.

#### 8.7.1. CF Architecture

CF networks utilize many distributed access points (APs) over a coverage area, significantly mitigating inter-cell interference and reducing transmission distances. For example, cell-free massive MIMO systems can deliver approximately a five-fold boost in 95%-likely per-user throughput compared to small-cell systems under uncorrelated shadow fading conditions, and a ten-fold improvement when shadow fading is correlated [87,88]. This design approach thus enhances the coverage probability and provides macro-diversity gains against large-scale fading, improving overall network performance by up to 30% [87,88,89,90,91,92,93]. These characteristics of CF systems can effectively mitigate severe path loss effects in NF-ISAC systems.

Moreover, CF-ISAC can overcome the drawbacks of single AP or BS-based conventional ISAC systems, such as obstructed observation angles in complex urban settings, which can reduce target detection accuracy [94,95,96,97]. In addition, CF-ISAC systems improve communication and sensing performance by coordinating distributed APs to execute both tasks. Thus, the diversity advantages of uncorrelated sensing observations at distributed APs/receivers will increase sensing performance. For instance, this coordination leads to an estimated 15–20% improvement in communication reliability and a 30% enhancement in sensing accuracy [94,95,96,97]. Despite its potential, only a few studies have characterized the performance of CF architecture in traditional ISAC systems thus far [94,95,96,97]. On the other hand, integrating CF and NF-ISAC will be an intriguing research opportunity for improving network performance.

CF network performance can be significantly degraded by impairments such as hardware imperfections in transceivers and SIC imperfections. One critical issue is the non-linearity of non-ideal RF front-ends [98]. These impairments impact reliability and security. Incorporating these impairments into NF-ISAC system models is crucial for analytical results to provide valuable insights into real-world conditions.

#### 8.7.2. BackCom

Due to the simple RF design, backscatter tags are cost-effective and have ultra-low energy requirements (a few nw-µW) [99,100,101,102,103]. They are typically equipped with sensors and perform passive reflection or backscattering rather than active transmission to convey data. Thus, they will facilitate future ambient-powered or battery-free (i.e., EH enabled) IoT networks and applications such as smart homes, cities, autonomous vehicles, industrial IoT, healthcare, and so forth [99,100,101,102]. These applications necessitate low-power communication and exceptional sensing capability. Thus, incorporating sensing into BackCom networks, also known as integrated sensing and BackCom (ISABC), enables IoT networks to extract crucial sensing/state information such as range, velocity, and angle for tracking, as well as environment learning and mapping [104]. EH is critical for BackCom, and non-linear models [105] could be used for design purposes.

On the other hand, as the backscatter devices are low-power EH devices with limited processing capabilities, their operational range and data rates are limited, i.e., the range is typically less than 6m and the data rate is less than 1bps/Hz [99,100,101,102]. Hence, many of these systems operate in the NF region of the transmitter/RF source [106,107,108,109]. However, except for a few works [106,107,108,109], the majority of the current literature focuses on FF BackCom systems. Nonetheless, almost all of these studies are measurement-based field trials designed to evaluate the NF impact of BackCom [106,107,108,109]. Only ref. [106] briefly explores RF sensing in the NF BC system, focusing on the interaction between NF signals and a shape-changing backscatter object, especially one with a low permittivity contrast to the background media. With minimal research on NF BC and no research on NF ISABC, there are plenty of opportunities to explore the feasibility of the NF BC and NF ISABC systems. Theoretical frameworks, in particular, must be established, since backscatter devices with sensing capabilities will enable identification, monitoring, and tracking in a wide range of future applications.

#### 8.7.3. RIS

Due to RIS’s energy and hardware efficiency in manipulating the wireless environment, RIS-integrated ISAC can unlock numerous new applications. Deploying RIS near the transmitter or receiver enhances passive beamforming gains for either interference management in ISAC or enhancing the performance of both communication and sensing [110]. For example, RIS can achieve over a 40% improvement in an ISAC system as compared to a no-RIS system [111]. Additionally, RISs operating at high frequencies can expand the NF region. Consequently, NF RIS-integrated ISAC is expected to become increasingly common [110].

Furthermore, the conventional RIS can only reflect signals that necessitate the users and the target to be at one side of BS and RIS [110]. To address this limitation, simultaneously transmitting and reflecting reconfigurable intelligent surfaces (STAR-RIS) have been developed, offering 360-degree coverage [68]. Consequently, STAR-RIS enhances the adaptability of network deployment with its full spatial coverage while optimizing signal propagation for both sensing targets and communication users. However, beamforming design and channel estimation are challenging in such systems, specifically as the number of RIS/STAR-RIS elements or BS antenna increases. Therefore, further in-depth research is needed to address these challenges.

## 9. Conclusions

In conclusion, the shift towards higher bandwidths and frequencies presents exciting opportunities for technologies like ISAC, facilitating applications that demand high accuracy and resolution. However, high-frequency systems combined with massive antenna arrays enlarge the NF region. Therefore, this work provides a comprehensive review and case study covering various aspects of NF and ISAC and their integration. Three different NF-ISAC case studies bolster this effort. Additionally, the existing challenges and open issues in this area are outlined. Finally, a broad perspective on future research directions is provided.

## Figures and Tables

**Figure 1 entropy-26-00773-f001:**
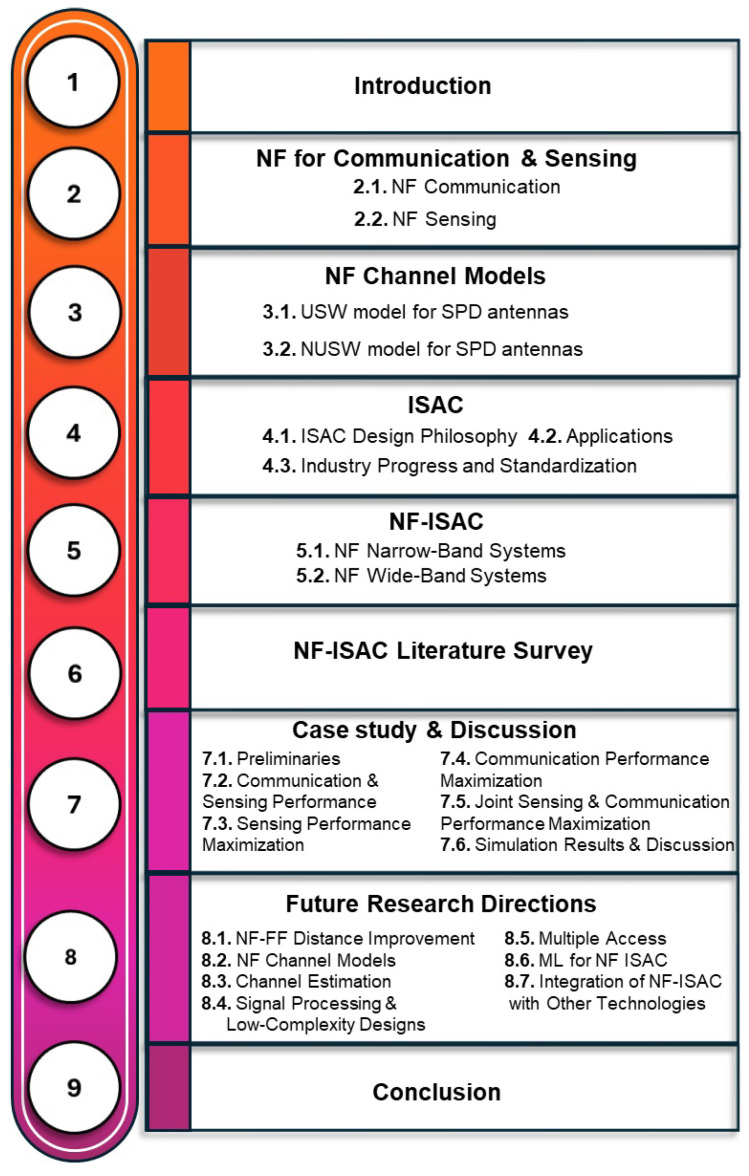
Outlook of this paper.

**Figure 2 entropy-26-00773-f002:**
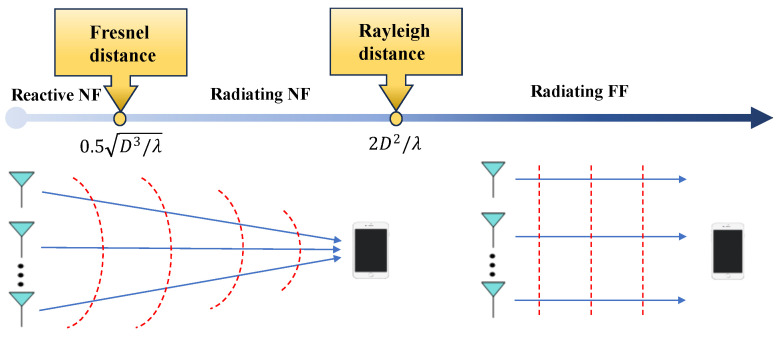
FF planar wavefront versus NF spherical wavefront.

**Figure 3 entropy-26-00773-f003:**
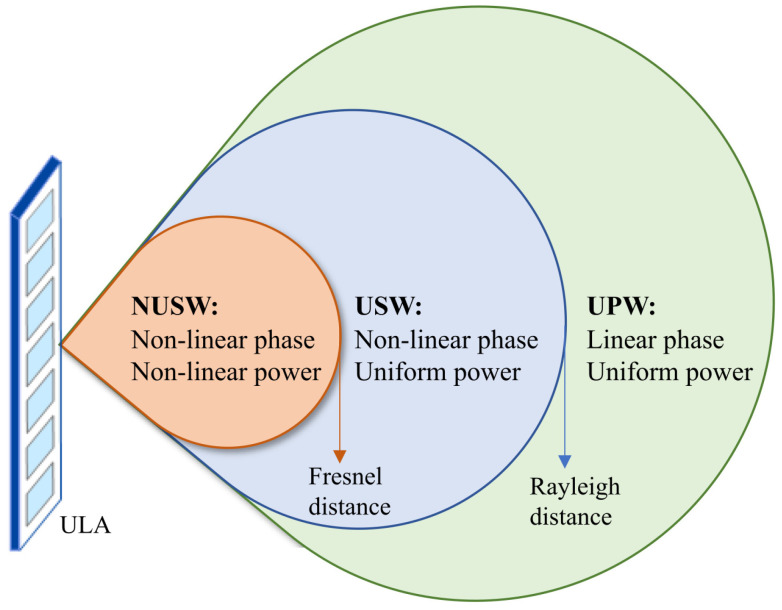
NF and FF separation.

**Figure 4 entropy-26-00773-f004:**
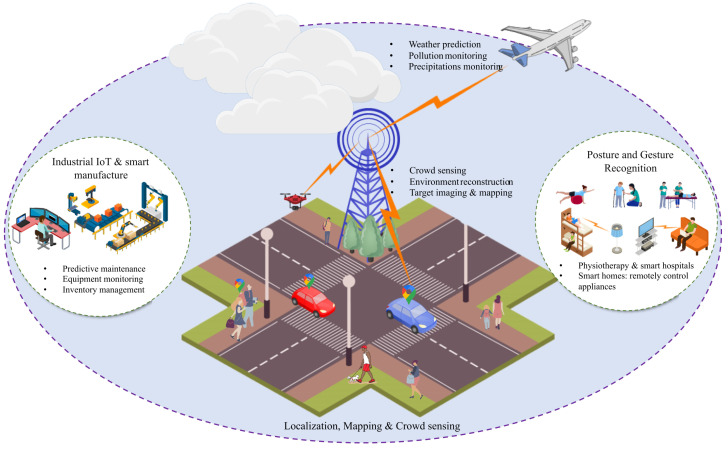
ISAC technology for future wireless networks.

**Figure 5 entropy-26-00773-f005:**
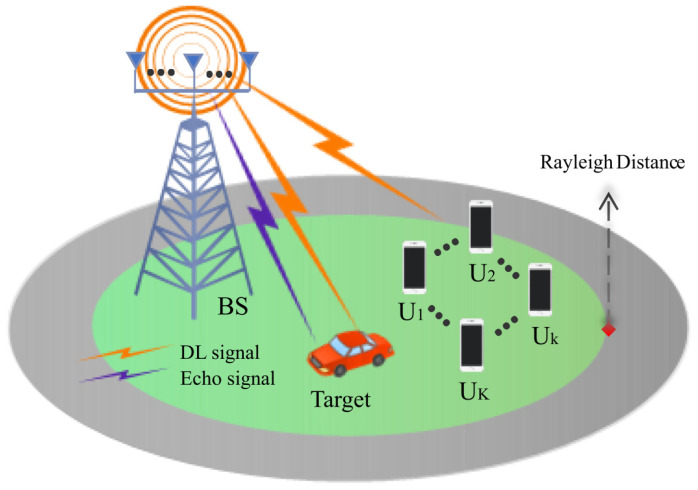
System model of DL NOMA empowered ISAC.

**Figure 6 entropy-26-00773-f006:**
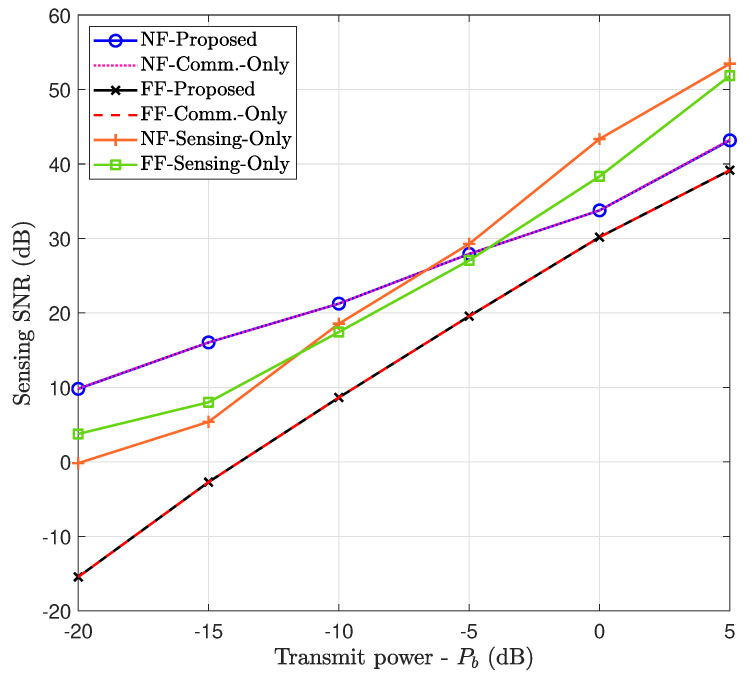
Case 1: Achieved sensing SNR versus BS transmit power.

**Figure 7 entropy-26-00773-f007:**
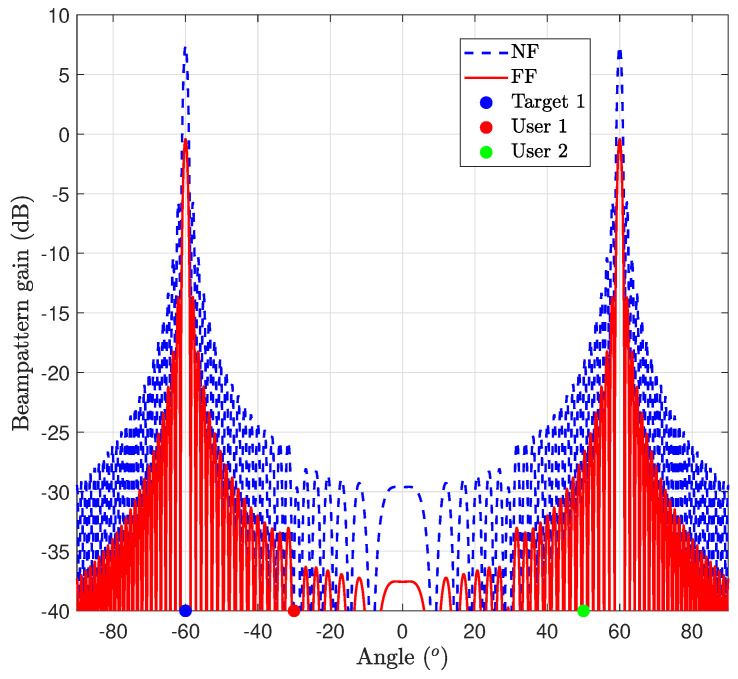
Case 1: Beam pattern gain.

**Figure 8 entropy-26-00773-f008:**
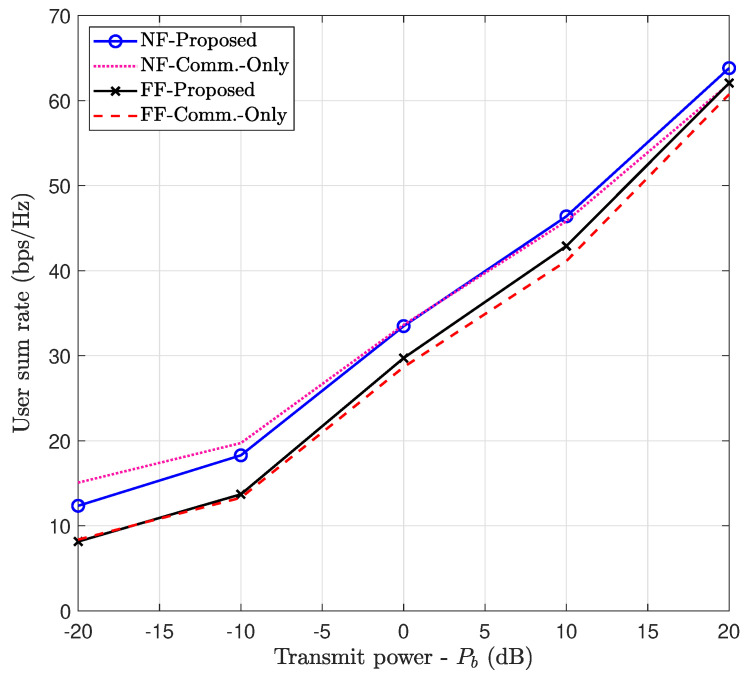
Case 2: Achieved user sum rate versus BS transmit power.

**Figure 9 entropy-26-00773-f009:**
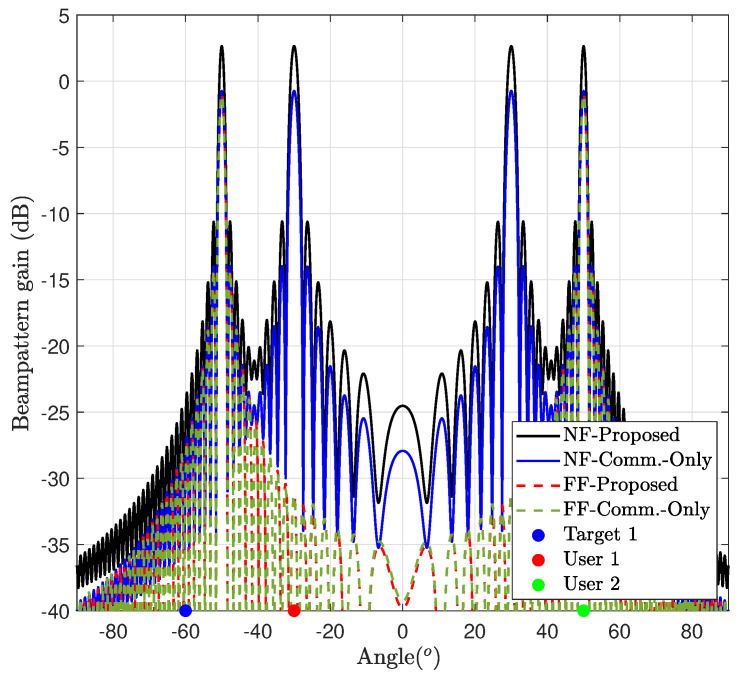
Case 2: Beam pattern gain.

**Figure 10 entropy-26-00773-f010:**
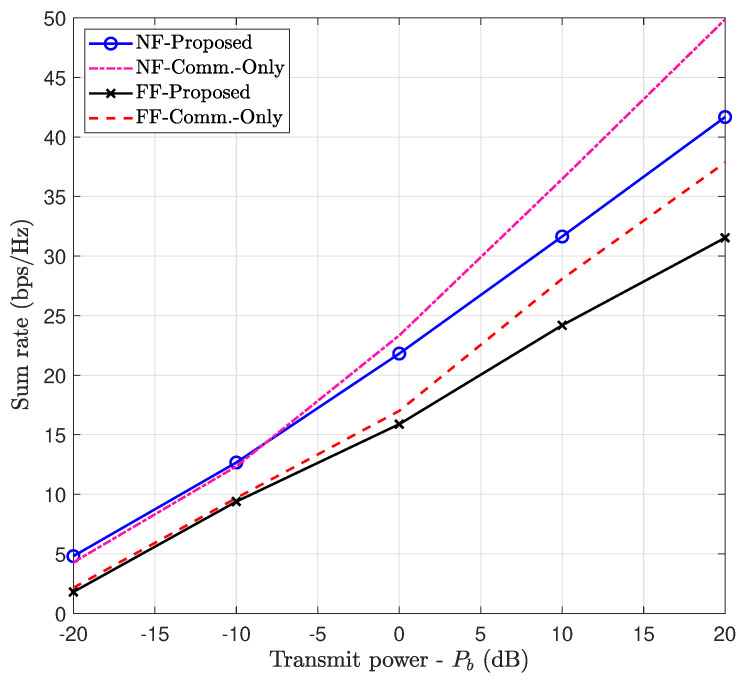
Case 3: Achieved user sum and sensing rate versus BS transmit power.

**Figure 11 entropy-26-00773-f011:**
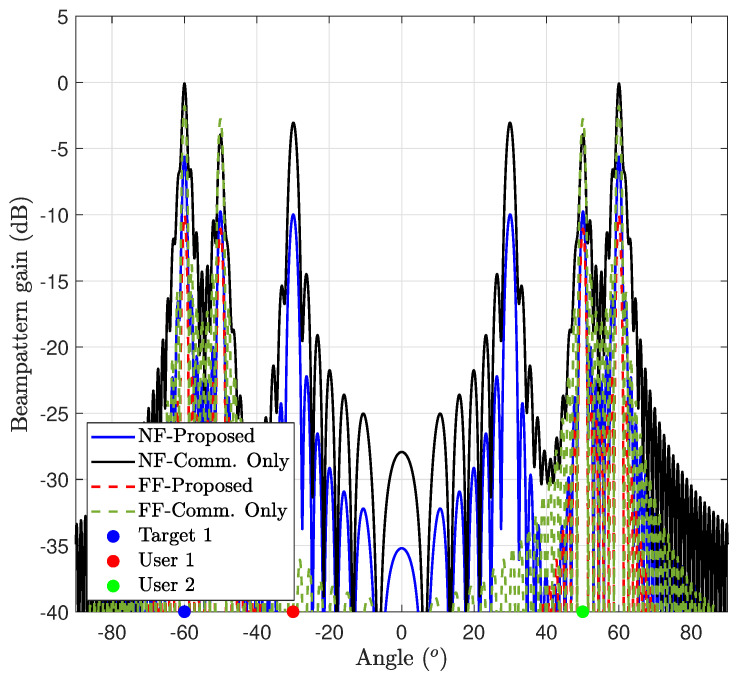
Case 3: Beam pattern gain.

## Data Availability

No new data were created or analyzed in this study. Data sharing is not applicable to this article.

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
