# Peer review of "A Roadmap for NF-ISAC in 6G: A Comprehensive Overview and Tutorial"

_entropy, 2024, doi:10.3390/e26090773_

Round 1

Reviewer 1 Report

Comments and Suggestions for Authors

The authors make a comprehensive review of near-field integrated sensing and communication. The subject is of interest in the context of future wireless network standards. Simulations are also performed to highlight aspects related to NF-ISAC. The paper is clearly written and presents useful results, but I think the type of the paper should be Review instead of Article. The authors should take into account the following comments and suggestions in order to improve the overall quality of the paper.

Comments regarding technical aspects:

-          In figure 2, it would be useful to explicitly mention the location of the transmitter;

-          It would be useful to mention in the context of sections 5.1 and 5.2 which of the application scenarios previously mentioned are matching one from the two cases;

-          Section 7 should be shifted towards the beginning of the paper;

Comments regarding editing aspects:

-          Some figures should be shifted before the text that is commenting them (Fig. 1, Fig. 3);

-          Better titles for subsections 6.6.1, 6.6.2 and 6.6.3 would be Sensing Performance Maximization (Case 1), Communication Performance Maximization (Case 2) and Joint Sensing and Performance Optimization (Case 3).

Comments on the Quality of English Language

Comments regarding grammar/typos:

-          …planar wavefront… instead of …planer wavefront… (row 187);

-          …there are… instead of …there is… (row 681);

-          …a single target is… instead of …the target is… (row 683).

Author Response

Thank you very much for your expert comments.  They have helped us to revise the manuscript, which has improved as a result.  Please find our detailed response in the attached PDF file. 

Best wishes!

C. Tellambura 

Reviewer 2 Report

Comments and Suggestions for Authors

The paper defines NF-ISAC 6G roadmap covering from a fundamental discussion to the potential applications. However, I would suggest the following to improve the paper:

1. The radio propagation channel and understanding are key to develop a successful NF-ISAC concept. I would suggest including more discussion on how the channel can be modelled theoretically and practically, what are key challenges and parameters are to be considered. Do we have reasonable models to capture NF propagation behaviour? 

2. The applications highlighted in this paper are very similar to what is available in most of the papers. I would suggest building discussion on what are unique areas/applications/use-cases in the NF-ISAC? FF is generally well established and a lot of hardware/software solutions have been developed over the years. It is important to discuss that what are major advantages so that the industry should follow or explore NF.

3. NOMA can certainly add another layer to accommodate multi-users as well established in FF, what new can NOMA bring in NF or it is just a question of repurposing the concepts from FF to NF.

4. How weather, rain and pollution predictions are linked in this work, very limited insight is provided. Does NF radar have better outcomes?

Comments on the Quality of English Language

N/A

Author Response

(The authors gave the same response as above.)

Round 2

Reviewer 2 Report

Comments and Suggestions for Authors

Thanks for the detailed response.

Comments on the Quality of English Language

N/A